# EMERGENT COORDINATION IN MULTI-AGENT LANGUAGE MODELS

**Christoph Riedl**
D'Amore-McKim School of Business,
Khoury College of Computer Sciences, and
Network Science Institute
Northeastern University
Boston, MA 02115, USA
`c.riedl@northeastern.edu`

## ABSTRACT

When are multi-agent LLM systems merely a collection of individual agents versus an integrated collective with higher-order structure? We introduce an information-theoretic framework to test—in a purely data-driven way—whether multi-agent systems show signs of higher-order structure. This information decomposition lets us measure whether dynamical emergence is present in multi-agent LLM systems, localize it, and distinguish spurious temporal coupling from performance-relevant cross-agent synergy. We implement a practical criterion and an emergence capacity criterion operationalized as partial information decomposition of time-delayed mutual information (TDMI). We apply our framework to experiments using a simple guessing game without direct agent communication and minimal group-level feedback with three randomized interventions. Groups in the control condition exhibit strong temporal synergy but little coordinated alignment across agents. Assigning a persona to each agent introduces stable identity-linked differentiation. Combining personas with an instruction to "think about what other agents might do" shows identity-linked differentiation and goal-directed complementarity across agents. Taken together, our framework establishes that multi-agent LLM systems can be steered with prompt design from mere aggregates to higher-order collectives. Our results are robust across emergence measures and entropy estimators, and not explained by coordination-free baselines or temporal dynamics alone. Without attributing human-like cognition to the agents, the patterns of interaction we observe mirror well-established principles of collective intelligence in human groups: effective performance requires both alignment on shared objectives and complementary contributions across members.

## 1 INTRODUCTION

Recent advances in generative AI (specifically LLMs) have led to tremendous advances in multi-agent systems (Qu et al., 2024; Hong et al., 2024; Qian et al., 2023; Li et al., 2024a; Subramaniam et al., 2025). Multi-agent systems often show impressive performance increases over single-agent solutions (Wu et al., 2023; Chen et al., 2023; Li et al., 2024b; Tao et al., 2024). A key argument behind such performance gains are claims to "greater-than-the-sum-of-its-parts" effects from differentiated agents (Chen et al., 2023, p.1). Connecting multiple differentiated agents has the potential benefit to leverage unique contributions that would not be available if the task was assigned to a single agent (Fazelpour & De-Arteaga, 2022; Tollefsen et al., 2013; Luppi et al., 2024; Lix et al., 2022). Conceptually, any group gains depend on synergy and emergence (Theiner, 2018; Riedl et al., 2021; Page, 2008; Hayek, 1945).[1] Despite impressive performance of many multi-agent systems we do not yet have a principled understanding when and how such synergy arises, what role agent differentiation plays, and how to steer it systematically. This all points to a crucial need for deeper understanding of collective intelligence in multi-agent systems, specifically whether multi-agent

---

[1]This is true for both human groups and multi-agent systems.

LLMs exhibit any synergy or complementarity at all—the necessary prerequisite for any absolute team-over-solo gains.

Determining whether multi-agent systems function as genuine collectives—rather than merely aggregating agents—requires a principled measure of synergy. We follow a purely data-driven approach to assess whether these systems exhibit higher-order synergy, characterized by structural coupling and joint information about future states and task outcomes, as evidence of emergent collective behavior. Intuitively, synergy refers to information about a target that a collection of variables provide only jointly but not individually (Rosas et al., 2020; Humphreys, 1997).[2] We develop a principled framework (Figure 1a) building on new insights in information theory (Rosas et al., 2020; Mediano et al., 2022b). Our goal is to develop methods to measure emergent role specialization, determine whether multi-agent systems show such signs of emergent synergy, explore whether synergy enables increased performance, and explore whether emergent behavior can be systematically steered with prompting.[3]

We apply our framework to study multi-agent systems of GPT-4.1, LLAMA-3.1-8B, LLAMA-3.1-70B, GEMINI 2.0 FLASH, and QWEN3 agents solving a simple group guessing task (Figure 1b) with three treatment interventions: a control condition, a condition that assigns a persona to each agent, and one that uses personas with an instruction to "think about what other agents might do" (a theory of mind (ToM) prompt). The paper is organized into addressing three concrete research questions:

RQ1: Do multi-agent LLM systems possess the capacity for emergence?

RQ2: What functional advantages—such as synergistic coordination and higher performance—arise when multi-agent systems exhibit emergence?

RQ3: Can we design prompts, roles, and reasoning structures that steer the internal coordination regimes of multi-agent to encourage outcome-relevant, goal-directed synergy?

Our findings provide evidence for multi-agent LLM system capacity for emergence and that it underpins performance. Through a variety of complementary analyses and robustness tests, we show that the coordination style across interventions is very different. While emergence is present in all, only the ToM-prompt condition leads to groups with identity-linked differentiation and goal-directed complementarity across agents: they operate as a dynamically stable, integrated, goal-directed unit. This is consistent with research on human groups showing that performance gains are not automatic (e.g., Stasser & Titus, 1985) and that complementarity needs to be balanced with integration and goal alignment (Theiner, 2018; DeChurch & Mesmer-Magnus, 2010; Riedl et al., 2021).

This paper helps us understand when and how multi-agent systems exhibit higher-order properties, their internal coordination structure, and how to control them. This can inform multi-agent system design by showing how to combine agents effectively. We make four contributions:

- Novel framework to quantify emergent properties in multi-agent systems based on information decomposition including conditional/residual variants and outcome-relevant partial information decomposition.

- Principled diagnostic approaches to localize where synergy resides and distinguish it from alternative explanations (such as heterogeneous learning rates). Specifically, we develop two surrogate null distribution tests (row-shuffle to probe identity-locked structure; column-shuffle for dynamic alignment). This design lets us tease apart "good" synergy aligned with task goals from spurious or misaligned synergy.

- Demonstrate how to steer emergence with specific prompts. Prompt-level manipulations causally change higher-order dependencies and reliably induces distinct coordination regimes, shifting collectives from spurious and misdirected synergy to stable and goal-aligned complementarity driven by differentiated identities.

- Internal coordination is measurable and controllable with interventions. Groups differ in variance, stability, and adaptability—properties relevant to reliability and deployment.

---

[2]The key claim explored in this paper is conditional, cross-agent synergy—i.e., synergistic information and coordinated differentiation across agents given the multi-agent constraint (not absolute outperformance over a solo agent). We do not attempt to establish team-over-solo superiority on this task.

[3]See GitHub for replication code `https://github.com/riedlc/AI-GBS`.

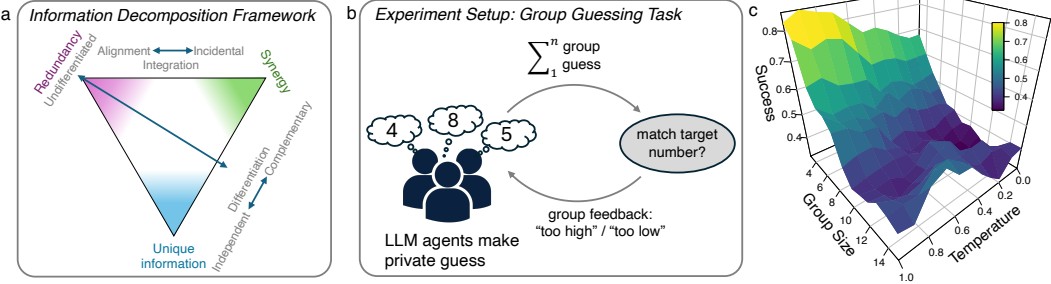

Figure 1: **a)** Information decomposition provides framework to explain tension in multi-agent systems. Agents are either undifferentiated or differentiated, provide independent or complementary information, which is either well aligned or incidental (adapted from Luppi et al., 2024). **b)** Experiment setup of the group binary search task. **c)** Preliminary experiments testing different group sizes and temperature settings. Surface values were smoothed using a local $3 \times 3$ weighted averaging filter, giving higher weight to each cell's original value to reduce noise while preserving local structure.

## 2 METHOD

**Group Task.** We study a group guessing game (developed by Goldstone et al. (2024) who called it "group binary search") without communication: agents propose integers whose sum needs to match a randomly generated hidden target number. Agents are unaware of each others' guesses and the size of their group, and receive only group-level feedback "too high" or "too low." The task is challenging because identical strategies induce oscillation and only complementary strategies yield success. This setting naturally pits redundancy (alignment) against synergy (useful diversity), and admits clear nulls via row-wise (break identities) and column-wise (break alignment) surrogates. The binary search setting is a minimalist testbed to isolate emergence of cross-agent complementarity and dynamic coordination, not to establish team-over-solo superiority. The implementation of a textbook-like XOR dynamic allows us to study the benefits of emergence capacity conditional on a multi-agent setup (and not to establish team-over-solo superiority). Research on humans has shown that groups can reliably solve this task through the emergence of specialized, complementary roles (Goldstone et al., 2024).

**Interventions.** Plain: The control condition uses only instructions for the guessing game, similar to those used in human subject experiments (Goldstone et al., 2024). Persona: For the persona condition, we follow recent insights into the use of personas within the LLM community (Chen et al., 2024) and ensure personas have relevant attributes: name, gender, age, occupation or skill background, pronouns, personality traits (Goldberg, 1992), and personal values (Cieciuch & Davidov, 2012). ToM: The ToM condition instructs agents to think about what other agents might do, and how their actions might affect the group outcome. This is akin to introducing social skills as defined in the human capital literature (Deming, 2022). See Appendix A.1 for all prompts, full persona generation recipes, and example persona.

**Analytical Framework.** How would we know if a multi-agent system shows emergent properties that could suggest that the sum is more than its parts? We develop a principled test based on a recent formal theory of dynamic emergence based on information decomposition (Rosas et al., 2020; Mediano et al., 2022b; Bedau & Humphreys, 2008). Our framework connects emergence with information about a system's temporal evolution—future states of the whole—with information that cannot be traced to the current state of its parts. This framework is based on partial information decomposition (PID; Williams & Beer, 2010) and time-delayed mutual information (TDMI; Luppi et al., 2022), is data-driven, quantifiable, and amenable to empirical testing with falsifiable conjectures against specific null hypotheses. Combining this with permutation tests, focused either on breaking connections between agents or connections across time allow us not only to quantity emergent properties but also to localize them in the system. This is crucial for multi-agent systems because we care about distinguishing synergy among specialized, differentiated agents from mere dynamic alignment.

We implement three tests. The first, termed *emergence capacity*, captures the ability of the multi-agent system to host *any* synergy. It measures the predictive synergy from two agents' current states to their *joint* future state, quantified using mutual information $I(\cdot; \cdot)$. For each pair of agents $(i, j)$ and time $t$, let the sources be $X_{i,t}$ and $X_{j,t}$, and define the bivariate target as the joint future state $t + \ell$, $T_{ij,t+\ell} \equiv (X_{i,t+\ell}, X_{j,t+\ell})$, where $\ell$ is the integration timescale (Mediano et al., 2022a). From the joint contingency table over $(X_{i,t}, X_{j,t}, T_{ij,t+\ell})$ we compute a two-source PID of the predictive information

$$I(\{X_{i,t}, X_{j,t}\}; T_{ij,t+\ell}) = \text{UI}_i + \text{UI}_j + \text{Red}_{ij} + \text{Syn}_{ij} \tag{1}$$

and take $\text{Syn}_{ij}$ as the pairwise dynamical synergy. A positive score $\text{Syn}_{ij} > 0$ indicates that the predictive information about the joint future is not recoverable from any single component. We compute this for all unordered pairs $(i, j)$ and take the median as group-level synergy capacity. This test is limited to detect synergy of order $k = 2$ but has the advantage that it does *not* rely on the definition of a suitable whole-system macro signal.

The second test, termed the *practical criterion*, concerns predicting a macro signal $V$ of the system. It asks whether the macro contains predictive information beyond any individual part. Given microstate $X_t = (X_{1,t}, \ldots, X_{n,t})$, macro $V_t = f(X_t)$. We align samples $(t, t + \ell)$ and compute the score as

$$S_{\text{macro}}(\ell) = I(V_t; V_{t+\ell}) - \sum_{k=1}^{n} I(X_{k,t}; V_{t+\ell}). \tag{2}$$

A positive value indicates that the macro's self-predictability exceeds what the sum of its parts can explain, thus indicating emergent dynamical synergy. We refer to "dynamical" synergy because targets are time-lagged and do not claim causal directionality beyond the temporal ordering (see Rosas et al., 2020, for a nuanced discussion). It is a coarse, order-agnostic screen sensitive to multi-part synergy (i.e., synergy of any order $\geq 2$). However, it is also penalized by redundancy across parts which can make the score negative even when higher-order synergy exists.

To explore system dynamics, we implement a *coalition test*, an extension of the practical criterion to triplets. Let $I_3$ be the mutual information between a triplet's current state and the future macro signal: $I_3 = I((X_{i,t}, X_{j,t}, X_{k,t}); V_{t+\ell})$. I.e., how much the three agents jointly predict the macro's future serves as a measure of the system's information-processing capacity (Mediano et al., 2022a). High $I_3$ indicates that agents' behaviors are tightly coupled and contain rich information about the emerging collective outcome, a pattern frequently associated with collective intelligence (De Vincenzo et al., 2017; Wicks et al., 2007; Mora & Bialek, 2011; Barnett et al., 2013). Low $I_3$ signals weak alignment, resulting in uncoordinated or chaotic collective behavior. Then

$$G_3 = I_3 - max(I_{2\{1,2\}}, I_{2\{1,3\}}, I_{2\{2,3\}}) \tag{3}$$

measures the additional predictive information the full triplet provides over the most predictive pair $I_2$ (a form of whole-minus-parts metric; Barrett & Seth, 2011; Mediano et al., 2025). If we find $G_3 > 0$ this means no pair is sufficient to capture information that the triplet contains about the macro signal at $t + \ell$. We again compute $I_3$ and $G_3$ for all possible triplets and take the median as group measure. This measure complements the other two by offering a coalition-level test geared towards functional relevance: it helps answers whether the joint information provided by coalitions of agents is actually about the goal. It allows us to localize where macro predictability depends on beyond-pair structure and to rule out "explained by best pair" explanations.

**Estimation Details.** As microstates $X_t = (X_{1,t}, \ldots, X_{n,t})$ we use each agent $i$'s guess at time $t$ and transform it into the deviation from equal-share contribution to hit the target number: $devs_{i,t} = raw_{i,t} - target/N$. This minimal transformation removes the trivial level differences imposed by the target and aligns agents on a comparable scale, so that remaining variation reflects coordination

(who compensates for whom). Over-/under-contributions are interpretable as the sign of deviations. As macro signal $V_t$ we take the group error $\sum raw_{i,t} - target$ (equivalently $\sum devs_{i,t}$).[4]

**Falsification Tests.** We perform two different falsification tests of each of the three criteria introduced above. Our primary test assesses significance by comparing the observed value against a null distribution computed on permuted (shuffled) instances of the *devs* data. We use two surrogates: row-wise shuffles (to break identities) and column-wise time-shift surrogates (preserve individual dynamics while disrupting cross-agent alignment). Full details in Appendix. We combine $p$-values across independently simulated groups (independent seeds, independently sampled targets, and no shared state) using Fisher's method. As a robustness test, we also compute bias-corrected (BC) estimates using time-demeaned data against a block-shuffled null with $\ell = 2$ to mitigate autocorrelation confounds. We use both (a) linear regression demeaning and (b) a functional baseline without between-agent synergy (see Appendix for details). These tests provide additional robustness to the $\ell = 2$ block size in the shuffled null and auto-correlation concerns.

**Entropy Estimation.** Small-sample entropy estimation is often challenging because the true distribution is only partially observed and many outcome categories receive few or even zero counts ("empty bins"). Such finite-sample estimation of information measures is biased upward and can yield spurious positive findings. Bias grows with more bins, dimensionality (e.g., using triplets instead of pairs), and small $N$ (few timesteps). The empirical plug-in estimator replaces true probabilities with sample frequencies and unseen (or under-sampled) outcomes often are assigned zero or very low probabilities (Hausser & Strimmer, 2009). To account for this issue, we follow best practices and take several steps. First, we use the Williams–Beer two-source PID (Williams & Beer, 2010) with the $I_{min}$ (minimum specific information) redundancy, estimated via plug-in probabilities. Second, we compute the emergence capacity and the coalition tests as order $k = 2$ instead of the entire system of $n$ agents. Second, we report $\ell = 1$ which is suitable to detect next-step oscillating behavior (see task description). Because trials end at success, episode lengths vary (see plot of rounds to success in Appendix). We report additional sensitivity tests using "early synergy" truncated to a fixed horizon of $H \in \{10, 15\}$ rounds to ensure comparability (results in Appendix A.2 and A.12; main analyses use full trajectories). Third, we apply quantile binning (with two bins) to reduce the dimensionality of our data (sensitivity analyses with three bins reported in A.8). Data are encoded as factors with fixed levels $1, 2$ (and for the joint target as the Cartesian product levels) to avoid dropping empty categories. Fourth, we use bias-corrected entropy estimator with Jeffreys' prior, which smooths estimates and avoids empty bins by adding $\alpha = \frac{1}{2}$ pseudo-counts in the Dirichlet estimator (Jeffreys, 1946). This reduces systematic inflation and cures zero-count pathologies. Finally, as a robustness check, we also report results using the Miller–Madow bias-corrected estimator (Miller, 1955), and the MMI redundancy measure (MMI typically overestimates redundancy, making synergy estimates more conservative; Mediano et al., 2025).

**Test of Agent Differentiation.** Our final pillar evaluates agent differentiation via hierarchical (mixed) modeling (Gelman & Hill, 2007), asking whether agents adopt consistent, person-specific patterns. We estimate a sequence of three models for each multi-agent experiment:

$$m_0: \quad y_i = \beta_0 + u_{\text{time}[i]} + \epsilon_i$$
$$m_1: \quad y_i = \beta_0 + u_{\text{time}[i]} + u_{\text{agent}[i]} + \epsilon_i$$
$$m_2: \quad y_i = \beta_0 + u_{\text{time}[i]} + u_{\text{agent}[i],0} + u_{\text{agent}[i],\text{time}[i]} + \epsilon_i$$

where $y_i$ are $devs_{i,t}$, $u_{\text{time}[i]}$ are random intercepts for (continuous) time, $u_{\text{agent}[i]}$ are agent-level random intercepts, and $u_{\text{agent}[i],\text{time}[i]}$ are agent-level random slopes varying by time. Using equal-share deviations removes target-level drift and makes agents directly comparable round-by-round. The random intercepts for time captures round-to-round shifts due to group-wide feedback and oscillation. Hence, agent effects reflect relative, identity-linked behavior rather than shared dynamics. The partial pooling of hierarchical models regularizes per-agent estimates, improving stability in short time series. We then use likelihood ratio tests to compare the nested hierarchical models. Comparing $m_0 \to m_1$ asks whether agents differ in their group contribution (some agents guess

---

[4]We also performed additional sensitivity analyses using alternative specification of the macro signal as first principle component of individual guesses as used in Rosas et al. (2020).

higher/lower than others), while $m_1 \rightarrow m_2$ tests whether agents vary in how much their contribution changes across rounds (learning rates). A significant $p$-values (e.g., below conventional 0.05 levels) indicates that the more complex model explains the data significantly better, suggesting the added random effects (e.g., agent-level differences or varying slopes) are meaningful in this group. This offers an interpretable, non-information theoretic test of monotonic learning-rate heterogeneity. Furthermore, this test does not require discretization, giving a complementary failure mode to the information-theoretic analyses.

Together, these four tests of our framework allow us to (a) detect higher-order structure in multi-agent systems, (b) assess whether it is driven by redundancy or synergy, (c) localize whether it is identity-locked vs. dynamic alignment, and (d) show whether the higher-order structure is functionally useful for the task. No single measure does all four. Combined with the prompt-level interventions, the framework allow us to test whether interventions (Plain, Persona, ToM) causally increases dynamic synergy, how it affects identity-locked differentiation, and goal alignment (i.e., increase in $I_3$).

# 3 EXPERIMENTS

## 3.1 PRELIMINARY EXPERIMENTS: GROUP SIZE AND TEMPERATURE

To see if LLMs can solve this task and how sensitive results are to different group sizes and temperature settings we ran a first set of preliminary experiments (see Appendix for prompt). We used OpenAI's `gpt-4.1-2025-04-14`. We varied group size from 3 to 15, and temperature from $[0, 1]$ in steps of 0.1. For each grid point we ran 50 group experiments (13 group sizes × 11 temperature settings × 50 groups = 7,150 experiments; Figure 1c). Multi-agent systems of GPT 4.1 agents can solve the task reliably but encounter substantial challenges. We fit a logistic regression, predicting success from group size and temperature. We find the task is significantly easier for smaller groups: each additional group member decreased the odds of success by roughly 8% (OR = 0.92, $p < 10^{-16}$), consistent with results as for human groups (Goldstone et al., 2024). Each unit increase in temperature increased the odds of success by about 50% (OR = 1.50, $p < 10^{-7}$).

## 3.2 MAIN EXPERIMENTS

For the main set of experiments, we chose groups of $N = 10$ because that appears to be the most difficult setup and a temperature of $T = 1$. We again used OpenAI's GPT-4.1 (version date 2025-04-14; OpenAI, 2025) with the prompts shown in Appendix A.1. We replicate each group experiment 200 times per treatment condition (600 experiments total). Overall success rate is not significantly different across the interventions (Figure 2a). We perform robustness tests using four other models: Meta's LLAMA-3.1-8B and LLAMA-3.1-70B-INSTRUCT (Meta, 2024), Google's `Gemini 2.0 flash` (DeepMind, 2025), and Alibaba's QWEN3 235B A22B INSTRUCT 2507 (Team, 2025). We query GPT-4.1 through the OpenAI API, QWEN throgh Cerebras, GEMINI through OpenRouter, and LLAMA through locally a hosted vLLM on a NVIDIA RTX Pro 6000 Blackwell.

# 4 RESULTS

## 4.1 EMERGENT SYNERGY

We first test if there are signs of emergence using the practical emergence criterion (predicting a time-lagged macroscopic group signal from micro-level states). First, we take the criterion computed using the equal-share contribution data and compare it against a null distribution of block-shuffled permutations. Individually, about 3.5% of experiments show a $p$−value below 0.05. A joint Fisher test using all $p$-values is highly significant, both overall and when tested within in each treatment condition separately ($p$-values are below $10^{-16}$). Our second test takes the bias-corrected versions of the measure and performs a Wilcoxon signed rank test of the null hypothesis that the median value is above 0 (Figure 2b). We report our most conservative results using time-trend demeaned data. The bias-corrected estimates are well above 0 in all conditions (Plain: $p = 1.5 \times 10^{-16}$; Persona: $p = 6.6 \times 10^{-7}$; ToM: $p = 0.02$). We report additional robustness tests using (a) raw

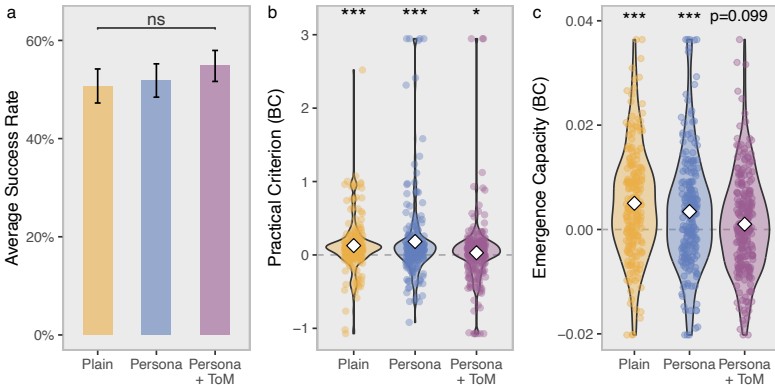

Figure 2: **a)** Group success across three interventions. **b)** Practical emergence criterion (bias corrected). **c)** Emergence capacity dynamical synergy (bias corrected). Data Winsorized at the 1st and 99th percentiles for visual clarity. Stars indicate significance level of Wilcoxon test. Notes: *** $p < 0.001$; ** $p < 0.01$; * $p < 0.05$.

data, (b) functional null model residuals, (c) using agent reactivity as data, and (d) bias correction using Jeffrey's smoothing and Miller-Madow in the Appendix.

We repeat analyses using the emergence capacity criterion. We find it is significantly above its null across conditions (Figure 2c); results hold under residualization and alternative entropy corrections (see Appendix for details). Taken together, across two different tests (one using a macro signal the other using future system behavior) and across a variety of robustness tests we find evidence of dynamic emergence capacity in multi-agent LLM systems (answering **RQ1**).

## 4.2 DYNAMICAL MECHANISMS

How is this emergence dynamically maintained and where is it located within the groups? We find time-trend demeaned bias-corrected $I_3$ around 0 in the **Plain** (Wilcoxon $p$-value 0.974) and **Persona** condition ($p = 0.846$) indicating groups that fail to escape chaotic, uncoordinated states. In contrast, the **ToM** shows significant positive mutual information ($p = 3.5 \times 10^{-14}$), collapsing the group's behavior into a predictable macro-state (Figure A2). The number of groups with significantly positive $I_3$ is substantially higher in the **ToM** condition (Figure 3a), consistent with enhanced collective intelligence. To formalize this transition dynamic, we compute the Total Stability of the system (computed as $I_3$ normalized by the entropy of the macro-signal) which serves as a proxy for the Lyapunov stability of the collective state (Haddad & Chellaboina, 2008). We find Total Stability indistinguishable form zero in the **Plain** (Wilcoxon $p$-value 0.976) and **Persona** ($p$-value = 0.858) indicating a "gaseous" state where groups fail to settle into a stable attractor. However, the **ToM** intervention induces a sharp increase in Total Stability ($p$-value = $2.9 \times 10^{-14}$). Theoretically, this suggests the **ToM** prompt acts as a control parameter, steering the multi-agent system from a chaotic regime into a deep basin of attraction where collective behavior stabilizes and information processing capacity spikes: turning the system from a collection of individuals into a collective group.

Decomposing this stability reveals the specific mechanism of coordination. While the system exhibits emergence capacity and practical emergence, we find the stability in group behavior is supported by pairwise alignment rather than irreducible triplet complexity. Specifically, triadic information gain ($G_3$) is around 0 in **Persona** and **ToM** (Figure 3c). The **Plain** conditions shows small positive $G_3 > 0$ (Wilcoxon $p$-value = 0.026) but given the near-zero Total Stability, this likely reflects transient stochastic correlations and oscillation rather than sustained coordination. In dynamical systems terms, the **ToM** prompt creates a deep basin of attraction where agents converge on a shared schematic (or Schelling point), making the collective state robust but informationally redundant. This mirrors "Mean Field" coupling in physics: because agents receive only global feedback and cannot observe individual peers, they couple to the aggregate signal rather than forming distinct local bonds. In such an environment, attempting complex higher-order synergy would be

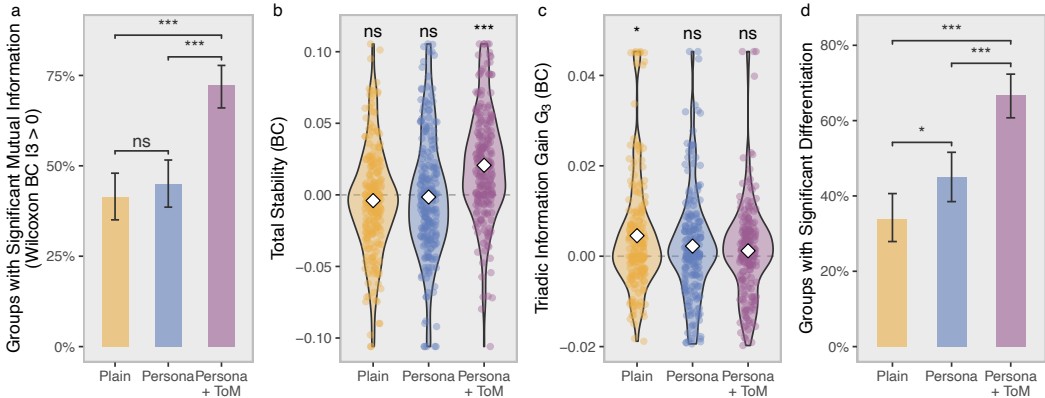

Figure 3: **a)** ToM-prompt condition has substantially more groups with significant $I_3$ content (above 0). **b)** Total Stability (time-delayed mutual information of triplets ($I_3$ normalized by macro-signal entropy, bias corrected). **c)** Information gain of triads over most informative dyad ($G_3$, bias corrected). **d)** Agent differentiation using hierarchical mixed model comparison (counting groups in which at least one test (different intercepts or slopes) is below $p < 0.05$. In panel a) and d), error bars show Wilson confidence intervals for binary data. Stars indicate significance level of test for equal proportion. Panel b) and c) shows bias corrected data with Jeffreys' prior. Data are Winsorized at the 1st and 99th percentiles for visual clarity. Stars indicate significance level of Wilcoxon test. Notes: *** $p < 0.001$; ** $p < 0.01$; * $p < 0.05$.

fragile; instead, the system converges on the efficient solution of dense pairwise alignment to the Mean Field.

Next, we explore whether agents evolve specialized roles and identities. Are there detectable between-agent differences in either the level of their contribution to the group guess or their temporal evolution (i.e., learning rate)? Using the hierarchical model comparison test we find that agents in many groups have differentiated identities (Figure 3a; a Fisher test for joint $p$-value evidence is again highly significant). There is substantially more differentiation among agents in the **Persona** condition, and even more in the **ToM** condition. The reasoning traces of agents often contain references to the "personal experience" of their assigned persona such as "In my experience (whether it's corraling cattle or wrangling numbers), starting toward the middle gives the group the best shot to zero in after the first feedback" (see Appendix for additional sample quotes). In Appendix A.10 we provide additional robustness tests ruling out heterogeneous learning rates as the sole source of differentiation and provide additional evidence for stable agent identities that persist over time. That is, in the **Plain** condition, differentiation results only from idiosyncratic noise and transient drift introduced by probabilistic LLM responses. With **Persona**, agents gain stable, identity linked behavioral preferences that refine their differentiation. The addition, **ToM** sharpens role differentiation through feedback loops from mutual predictive modeling by conditioning on the public history (which functions as a common ground and coordination device; Lewis, 1969; Skyrms, 2010). This converts small persona-induced asymmetries into stable, self-reinforcing roles, creating the basis for strategic complementarity and stable alignment (answering **RQ2** and **RQ3**)

### 4.3 FUNCTIONAL ROLE OF EMERGENCE IN GROUP PERFORMANCE

What functional differences—if any—does emergence allow? We use regression analysis to explore the joint effect of synergy and redundancy (based on emergence capacity formulation) while carefully controlling for endogeneity of variables measuring emergent properties and performance and controlling for sample selection bias with stabilized inverse probability weighting (see Appendix A.11 for method details). On their own, higher levels of either synergy or redundancy do not predict success. However, when both are present, performance improves significantly (significant interaction with $\beta = 0.24$; $p = 0.014$). In marginal-effect terms, redundancy amplifies the benefit of synergy on the log-odds scale by 27%; and vice versa synergy amplifies benefits of redundancy by 27%. This pattern implies that systems benefit when redundant pathways create goal alignment,

while synergistic interactions extract novel, non-overlapping information—together enabling higher overall performance. We complement the regression analysis with causal mediation analysis (Imai et al., 2010). While reaching only marginal significant levels the effect is consistent: the ToM treatment causally increases performance indirectly by increasing synergy (ACME $= 0.034$ [95%CI: $-0.000 - 0.07$], $p = 0.053$). This aligns with the interpretation that performance benefits emerge when systems achieve both redundancy (aligned toward a common goal) and synergistic integration (differentiated, complementary roles)—a form of functional and collective complexity (Varley et al., 2023; Luppi et al., 2024; Sterelny, 2007).

### 4.4 EXPERIMENTS USING OTHER MODELS

To assess generalizability of our findings, we repeat experiments with four other models: LLAMA 3.1 8B and 70B, GEMINI 2.0 FLASH, and the reasoning model QWEN3 (see Appendix A.13 for full details). We find that the capacity for emergent synergy is robust across high-capability models: LLAMA 70B, GEMINI, and QWEN3 achieved success rates on par with GPT 4.1 and exhibited strong evidence of emergence using both the emergence capacity and practical criteria. Consistent with our main results, these high-capability models achieved higher success rates, more differentiation, and stronger evidence of emergence in the Persona and ToM conditions compared to the Plain baseline. Conversely, the smaller LLAMA 8B largely failed to break oscillatory cycles and develop goal-directed cross-agent synergy due to its lower ToM reasoning reasoning capacity (Xiao et al., 2025). Crucially, our analysis of the reasoning model QWEN3 uncovered a distinct structural challenge we term *paralysis under coordination ambiguity*. QWEN3 agents enter infinite chain-of-thought loops when attempting to reconcile local binary search strategies with noisy group feedback. We provide a detailed taxonomy of these reasoning failure modes—including mutual mental modeling traps—in Appendix A.13, highlighting a critical frontier for research of reasoning models in multi-agent systems (Piedrahita et al., 2025).

## 5 RELATED WORK

Shortly after the release of ChatGPT, the community began exploring multi-agent LLM systems, where multiple LLM agents interact with each other. One early example is Park et al. (2023), who simulated a small population of "individuals" in a *The Sims*-style environment with friends, houses, and jobs, showing the emergence of rich social behaviors such as coordinating a Valentine's Day party. This was followed by multi-agent systems for complex tasks such as software development (Chen et al., 2023; Hong et al., 2024; Qian et al., 2023), healthcare (Li et al., 2024a), and other domains (Subramaniam et al., 2025; Ashery et al., 2025). Federated systems or "society of models" also gained attention (Juneja et al., 2024; Li et al., 2024b). These systems often achieve significant performance gains over single-agent baselines (Wu et al., 2023; Chen et al., 2023; Li et al., 2024b; Tao et al., 2024).

A recurring explanation for such gains invokes "greater-than-the-sum-of-its-parts" effects (Chen et al., 2023, p.1), often attributed to division of labor among differentiated agents (Subramaniam et al., 2025). Yet while this work is "inspired by human group dynamics" (Chen et al., 2023, p.1), it rarely evaluates whether the same principles that underlie effective (and ineffective) human groups also emerge in multi-agent systems. Two gaps stand out. First, human groups often coordinate by developing *role specialization* (Goldstone et al., 2024). Developers of multi-agent systems often intuitively induce such role specialization (such as "programmer", "tester", and "CEO"; Qian et al., 2023) yet have not systematically evaluated the impact on emergent shared cognition beyond win-rate comparisons. Second, groups only outperform individuals if their members contribute different and complementary information to the group task (Theiner, 2013; DeChurch & Mesmer-Magnus, 2010; Fazelpour & De-Arteaga, 2022; Nickerson & Zenger, 2004). Effective groups balance complementarity—members contributing distinct information—with redundancy, the alignment on shared goals (Theiner, 2013; DeChurch & Mesmer-Magnus, 2010; Fazelpour & De-Arteaga, 2022; Luppi et al., 2024; Tollefsen et al., 2013). Too much of either undermines performance. Unlike win-rate–centric evaluations—and tests of whether agents in multi-agent systems can sustain cooperation per se (Piedrahita et al., 2025; Piatti et al., 2024)—we propose falsification tests and null modeling, thus extending multi-agent frameworks with formal theory. While these principles likely apply to LLM-based collectives given the universality of the integration-

segregation tradeoff (Tononi et al., 1994), it remains unclear whether agents develop differentiated roles, whether such roles complement each other, how to steer it with prompts, and what role ToM capacity of models plays for collaboration (Westby & Riedl, 2023; Kleiman-Weiner et al., 2025; Riedl & Weidmann, 2025).

# 6 CONCLUSION

Collectives of LLM agents are emerging as a powerful new paradigm, capable of tackling tasks that exceed the reach of any single model. Yet raw capability alone may not determine their effectiveness. Effective multi-agent systems depend on acting as integrated, cohesive units as opposed to loose aggregations of individual agents. In collective intelligence, this distinction often separates groups that merely average their parts from those that achieve true synergy (Riedl et al., 2021). This paper asks whether such coordination dynamics also matter for LLM collectives, what functional benefits they have, and whether we can design prompts, roles, or reasoning structures that foster such integrated yet synergistic behavior. The framework we developed connects human group cognition theory to LLM multi-agent systems, offering a novel conceptual bridge. Methodologically, we demonstrate how to operationalize group-level synergy in AI collectives using quantitative information-theoretic measures. We also provide specific design principles for creating productive LLM collectives that are aligned on "shared goals," which can inform multi-agent orchestration tools, cooperative AI, and mixed human–AI teamwork. We further ground our information-theoretic framework in dynamical systems and game theory, showing that theory-of-mind interventions act as a control parameter that shifts multi-agent interactions from disordered to stable regimes with goal alignment. This work extends ToM research beyond standard false-belief tests to realistic collaborative tasks (Shapira et al., 2024; 2023) and points to ToM-related failure modes in reasoning models. Those insights will help us understand when multiple LLMs working together will be beneficial, why that is, and inform their design.

We note that evidence of higher-order synergy should not be interpreted as implying sophisticated cognition or consciousness. As conceptualized here, synergy is a structural property of part-whole relationships within multi-agent interactions. Such higher-order structures are known to arise in simple systems, including agents governed by reinforcement learning (Fulker et al., 2024) or via basic nonlinearities in contagion processes (Iacopini et al., 2019; Lee et al., 2025). Without attributing human-like cognition to the agents, the patterns of interaction we observe mirror well-established principles of collective intelligence in human groups (Riedl et al., 2021; DeChurch & Mesmer-Magnus, 2010): effective multi-agent performance requires both alignment on shared objectives and complementary contributions across members.

**Limitations and Future Work.** This work focuses on developing a framework, characterizing emergence, and localizing it. While we explore the degree of goal alignment via a performance-related macro signal and performance effects of "early synergy", the analysis linking synergy and redundancy to performance is challenging because synergy and redundancy are often co-dependent and emerge alongside performance. Future work should aim to more directly connect measures of synergy with performance. Despite a significant number of robustness tests, estimating entropy and establishing synergy is difficult. Furthermore, this work established results only using a single task—albeit one that is uniquely suited to studying synergy given how closely it mirrors ideal case of complementary behavior in a minimalist setting. More work will be necessary to convincingly establish when and how multi-agent LLM systems develop emergent synergy across tasks and measures. Given the small-data setting some of our information theoretic measures are limited to computing dynamic emergence on order $k = 2$ which is bound to miss synergy of a higher order. Finally, this work points to the importance of ToM. LLMs capacity to act in a ToM-like manner appears crucial to achieve functional alignment in multi-agent systems. This should provide additional motivation for research on ToM in LLMs.

## ACKNOWLEDGMENTS

We thank Andrea Barolo for some early work on the implementation and Amritesh Anand for excellent research assistance on the project. We thank Fernando Rosas, Yonatan Belinkov, Patrick Forber, Koyena Pal, and David Bau for comments. This work was supported by the D'Amore-McKim School of Business' DASH initiative which provided access to computational resources.

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

# A APPENDIX

## A.1 PROMPTS

---

**Preliminary Experiments**

You are playing a sum guessing game. Your goal is to help your group sum to the mystery number. Your guess range is 0 to 50.
Game History:
Round 1: Your guess: 25
Result: too HIGH
Round 2: Your guess: 12
Result: too HIGH
...
Based on this feedback, what should your next guess be?
Respond with only an integer between 0 and 50.

---

**Plain**

You are playing a sum guessing game. Your goal is to help your group sum to the mystery number. Your guess range is 0 to 50.
Game History:
Round 1: Your guess: 25
Result: too HIGH
Round 2: Your guess: 12
Result: too HIGH
...
What is your guess this round? Always start with the efficient strategy in guessing games which is to use a binary search approach: guessing the midpoint of the current range. Always anchor your guess on the group feedback from previous rounds (too high / too low).
End your answer with: FINAL GUESS: [0-50]

---

**Persona**

[PERSONA – Sample: You are Andrej, the Quantum Computing Engineer: Andrej, a Serbian engineer in Berlin, is one of a handful of experts in cutting-edge quantum tech. Systematic, inquisitive, and fond of classical music, Andrej also volunteers as a coding mentor for refugee youth.]
You are playing a sum guessing game. Your goal is to help your group sum to the mystery number. Your guess range is 0 to 50.
Game History:
[GAME HISTORY]
What is your guess this round? Always start with the efficient strategy in guessing games which is to use a binary search approach: guessing the midpoint of the current range. Always anchor your guess on the group feedback from previous rounds (too high / too low).
End your answer with: FINAL GUESS: [0-50]

---

---

**ToM**

[PERSONA]
You are playing a sum guessing game. Your goal is to help your group sum to the mystery number. Your guess range is 0 to 50.
Game History:
[GAME HISTORY]
What is your guess this round? Always start with the efficient strategy in guessing games which is to use a binary search approach: guessing the midpoint of the current range. Only as a secondary approach, carefully think through step-by-step what others might guess and how the contributions of others contribute to the sum of the group guesses for the mystery number. Consider what roles other agents might be playing (e.g., guessing higher or lower) and adapt your own adjustment to complement the group. Always anchor your guess on the group feedback from previous rounds (too high / too low).
End your answer with: FINAL GUESS: [0-50]

---

To generate personas, we used the following prompt with `gpt-4.1-2025-04-14`.

---

**Persona Generation**

Generate a list of 20 personas. Each persona description should be about one paragraph long. Personas should include diverse people with different backgrounds, jobs, skills, different preferences, and different personalities. Format the output as follows "You are [name], the [job title or professional identity]: [name] [rest of the persona]."

---

## A.2 ROUNDS TO SUCCESS

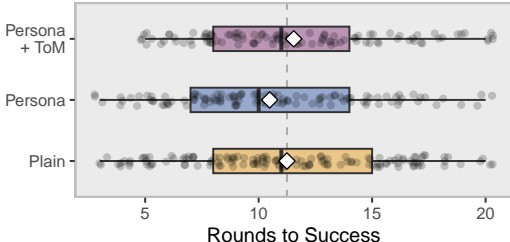

Figure A1: Conditional on success, we plot which round it was achieved.

## A.3 SAMPLE QUOTES FROM REASONING TRACES

Below we show some illustrative quotes to illustrate both diversity and reasoning about others. Clear reasoning about other agents in the ToM condition with different personal approaches (from same group, in round 3, with identical group feedback too high, too high, too low:

> Because it's possible others might go for 4 or 5 (the absolute lower bound or just above the last "too low"), and someone else might go for 7 or 8, I stick with the most efficient: 6.

> I'll round down to stay cautious (as sometimes others might guess the midpoint up), so I'll guess 8.

> Now, before some revolutionary comes in and throws a wild number for the sake of dramatic effect ("Viva el ocho!"), it's sensible for me to go with the midpoint. If my compañeros are clever, they'll flank me with 8 and 10, and we'll box in this mystery number tighter than a late-night metro in Gràcia. So, unless you want me to make a "radical" guess for the sake of satire (which, let's be honest, sometimes works), I'll go boring and efficient.

> If anyone else in the group is feeling feisty and picks 9 or 10, my 8 will help cover the lower part safely.

### A.4 FALSIFICATION TESTS: ADDITIONAL METHOD DETAILS

We perform a falsification test of each of the three criteria introduced above in two different ways. In our robustness test, we remove time trends from the equal-share deviations via agent-level linear regression residualization to mitigate autocorrelation confounds. We then generate a null distribution of our statistics computed on column-wise block shuffled data (we use $B = 200$ random shuffles for main analyses and confirm results using $B = 1,000$ on the emergence criteria tests). We permute time indices jointly across agents in blocks of length $\ell$, preserving within-block identity alignment while disrupting cross-agent temporal coupling beyond short time frames. Finally, we compute the bias-corrected estimate as the observed value minus the median of the null. We then perform a Wilcoxon signed-rank tests to test whether the bias-corrected values exceed zero ($p(H1 > 0)$. In some of our sensitivity tests we also perform a full within-row shuffle completely permuting guesses within each trial across agents (breaking agent identities), while preserving row constraints. This role differentiation or identity-specific coordination, including time-trend effects tests identity-locked differentiation beyond task induced temporal dynamics. Comparison between results using the full row-shuffle vs. the column-block-shuffle separates signs of identity-locked (specialized roles) from dynamic alignment (mutual adaptation, turn-taking, oscillation). In addition to regression residualizing we also implement a functional baseline (see Appendix).

### A.5 EMERGENCE CAPACITY: FULL RESULT DETAILS

Next, we test if there are signs of emergence capacity, again using both the $p$-value test on raw data and the test whether bias corrected values are above 0. We compute the dynamical synergy, which measures how much of the system's future behavior is only predictable from the whole system and not any subset of its parts. The main test finds that about 32% of groups show significant emergence capacity with $p$-values below 0.05 (**Plain**: 37%; **Persona**: 44%; **ToM** 18%). A joint Fisher test of all $p$-values is also highly significant. As robustness test we check whether the bias corrected value of dynamical synergy is above 0 using the Wilcoxon test (Figure 2b). The test is highly significant overall, as well as in the **Plain** and **Persona** condition. The **ToM** condition is marginally significant ($p = 0.099$). See analyses using $K = 3$ bins for additional discussion and evidence of significant synergy in the **ToM** condition.

### A.6 ROBUSTNESS TESTS USING EMERGENCE CAPACITY CRITERION

Using MMI PID with Miller-Madow bias correction, we find 20% of $p$-values are below 0.05 (across treatment conditions: 0.22%, 26%, 12.4%). Fisher test is highly significant overall, and in each condition individually (all $p = 0$).

### A.7 ROBUSTNESS TESTS USING PRACTICAL EMERGENCE CRITERION

The main paper outlines how we use regression to residualize data from time trends. Since this removes only a specific version of time trend (in this case linear) it may not capture the specific time trend present in the data. We creating a baseline expectation for behavior under a null scenario (no synergies) and then measuring deviations from that baseline. Specifically we implement a naive agent that performs deterministic binary search. Groups of this null-model agent are generally unable to solve the task and oscillate between guessing too high and too low (they only solve it when the target number happens to be divisible by the group size). This provides a residualization against a generative null model correction (aka a functional baseline).

### A.8 ROBUSTNESS TESTS USING THREE BINS

Quantile binning with $K = 2$ could potentially compress dynamics too much, blur role structure, and inflate apparent synergy by introducing threshold artifacts. We therefore report sensitivity analyses

|  | Raw Data (BC) | Time-Trend Residualized (BC) | Residualized against Functional Null (BC) | Reactivity (BC) |
|---|---|---|---|---|
| Min. | -4.234 | -2.476 | -3.323 | -2.650 |
| 1st Qu. | 0.105 | -0.041 | 0.002 | -0.015 |
| Median | 0.187 | 0.073 | 0.110 | 0.111 |
| Mean | 0.352 | 0.114 | 0.183 | 0.313 |
| 3rd Qu. | 0.406 | 0.168 | 0.269 | 0.352 |
| Max. | 4.881 | 4.881 | 3.886 | 4.869 |
| Wilcoxon $p$ (H1: $> 0$) | 0.000 | 0.000 | 0.000 | 0.000 |

Table A1: Robustness tests using alternative version of the practical emergence criterion.

using $K = 3$ bins for our main analyses. We find consistent results and support for the presence of emergence using three bins (instead of two as in the main analyses).

**Emergence Capacity.** 20% of individual experiment $p$-values are below 0.05 and Fisher test is highly significant (the largest $p$ value is in the **ToM** condition with $4.47 \times 10^{-3}$ the others are much smaller). Wilcoxon signed rank test for $\mu > 0$ of the bias-corrected and time-trend demeand data are as follows. Overall: $p = 0.005$; **Plain**: $p = 0.0004$; **Persona**: $p = 3.2 \times 10^{-6}$; **ToM**: $p = 0.991$. Wilcoxon signed rank test for $\mu > 0$ of the bias-corrected and functional-null demeand data are as follows. Overall: $p = 0.948$; **Plain**: $p = 0.999$; **Persona**: $p = 0.896$; **ToM**: $p = 0.025$. Using this stricter null indicates that only the **ToM** condition exhibits complementary cross-agent structure not explained by shared trends or independent feedback-following. That is, the stricter test isolates the construct of interest—synergy beyond coordination-free dynamics—and **ToM** is the only condition that passes it. The high $p$-value for **ToM** under the simpler time-trend demeaning should be taken as a methodological sensitivity (sparsity and a fat null). Together they show the **ToM** effect is robust to deconfounding but fragile under known low-power discretization. This is also in line with the stricter early synergy tests presented in Section A.12 which are also only passed by the **ToM** condition.

**Practical Criterion.** 5.8% of individual experiment $p$-values are below 0.05 and Fisher test is highly significant ($p < 10^{-16}$). Wilcoxon signed rank test for $\mu > 0$ of the bias-corrected and time-trend demeand data are as follows. Overall: $p = 3.6 \times 10^{-8}$; **Plain**: $p = 3.1 \times 10^{-5}$; **Persona**: $p = 0.001$; **ToM**: $p = 0.009$.

## A.9 ADDITIONAL RESULTS ON DYNAMICAL MECHANISM

The main paper shows results for Total Stability: $I_3$ normalized by the entropy of the macro signal. Here we show results for $I_3$ as well (Figure A2). They are nearly identical given that the entropy of macro signal if $\approx 1$ given quantile binning with two bins.

## A.10 ARE IDENTITIES PERSISTENT?

If some agents learn faster than others, even without persistent "roles," synergy can emerge simply because differentiated trajectories create complementarity across time. As indeed the mixed model test shows that 32% of groups have significant slopes provides strong evidence for the presence of heterogeneous learning rates. That is, heterogeneous learning rates could be a confound explaining some of our observed synergy effects. To rule out this alternative explanation we explore if agents have *persistent* "identities." We find the block-shuffle nulls show more significant $G_3$ than the full row shuffle. This suggests that synergy depends on temporal continuity of identities (not just on different learning rates at each timepoint). Furthermore, the hierarchical model test shows significant variance even without varying slopes, indicating stable role-like differences beyond learning rates. Finally, our analyses removing time trends and the functional null-model also remove learning rate slopes (Table A1). The functional null-model in particular does this in a non-linear way that is directly tied to our task. In summary, we find both heterogeneous learning rates and stable identity-linked differentiation. Heterogeneous learning rates alone cannot explain the observed emergent synergy that persist across time (providing additional insights regarding **RQ2**).

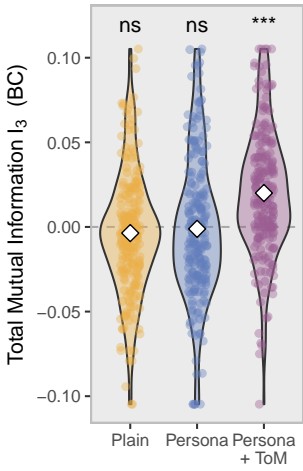

Figure A2: $I_3$ mutual information (time-trend demean, bias corrected) with Wilcoxon $> 0$ $p$-value indicators.

Together, the results suggest a nuanced picture of emergence in multi-agent systems. **Persona** and **ToM** produce agents with distinct, stable identities—moving agents on the differentiated-undifferentiated axis in Figure 1a. This differentiation shapes the space in which other agents can adapt, and specialization by one agent constrains the (useful) actions of the others (agents mutually constrain each other; cf. Tollefsen et al., 2013)—shifts on the interdependent-complementary axis. When combined with **ToM**, mutual adaptation both strengthens agent differentiation (creating the foundation for possible synergistic complementarity) while also increasing mutual alignment on shared goals (the target signal)—shifts on the integration axis. Differentiation enables synergy, but without redundancy and integration, synergy alone does not translate into higher collective performance. Agents in the **ToM** condition develop specialized, complementary roles while achieving stronger shared goal alignment: they are simultaneously more differentiated yet also more integrated along highly overlapping task-relevant information which the capacity for synergy can now exploit in productive ways.

### A.11 ADDITIONAL DETAILS FOR PERFORMANCE ANALYSIS

What does synergy enable? In particular, does synergy affect performance? Connecting emergent properties of synergy and redundancy to performance is a challenging task for two main reasons. First, redundancy and synergy sit at opposite ends of the informational spectrum of correlations. Systems constrained by limited resources (such as energy or bandwidth) cannot maximize both simultaneously. The "informational budget" of correlations can be spent on shared overlap (redundancy) or on higher-order joint structures (synergy), but not both simultaneously (Williams & Beer, 2010; Varley et al., 2023; Luppi et al., 2024). Second, emergent properties are dichromatic—they require time to evolve (Bedau, 2002)—making them endogenous with run-length of the simulation. Namely, simulations in which the multi-agent system performs worse and takes longer to solve the task (or fails to solve the task altogether) provide them with more time to evolve sophisticated group coordination patterns; while successful runs terminate early, giving collectives less time to evolve emergent properties. This complicates meaningful analyses as it can make it appear that higher synergy is correlated with failure. In technical terms: time is an endogenous variable and confounds both performance and observed values of synergy and redundancy.

We address these issues as follows. First, we focus our analysis on the interaction of synergy and redundancy, rather than each variable individually to capture the tradeoff between the two. Second, we compute both emergent properties (synergy and redundancy) only on early rounds (in our case, 10 time steps). We chose 10 rounds as a good tradeoff between giving the multi-agent systems enough time to evolve, while limiting the amount of runs that have already completed. While this reduces our ability to detect the full emergent potential of the multi-agent system (since agents have less time to evolve) it creates a clean apples-to-apples comparison between measured values (they are all

based on exactly 10 time steps). In that sense, our analyses likely underestimate the full effect of synergy because we only capture the early onset of synergy. Focusing on only the first 10 time steps, leads to additional complications, however. About 23% of experiments finish early and do not make it till round ten (roughly balanced across treatments with 22% in Plain, 25% in Persona, and 20% in ToM). Simply removing these observations would bias our results since this censoring occurs post-treatment assignment: removing those observations would introduce sampling bias. Instead, we take these observations as censored (i.e., we cannot compute the early synergy measure) and control for this censoring using stabilized inverse probability weighting (IPW; Wooldridge, 2002). That is, in a separate modeling step we predict (via logistic regression and only task difficulty as exogenous predictor variables that is independent of the treatment) the likelihood of observing (early) synergy and redundancy.[5] We then take the inverse of the predicted likelihood as weight in the causal mediation analysis. This adjust observations in the primary analysis with the inverse of the probability of making such observations.

Our first analysis is a plain quasibinomial regression on the binary success vs. failure outcome[6] with *synergy*, *redundancy*, their interaction, task difficulty as controls, and dummy indicators for the three treatment conditions. We use our most conservative estimate of synergy and redundancy with the bias-corrected estimates with high null model sampling ($B = 2,000$), and the conservative MMI redundancy (Mediano et al., 2025) with MI estimated with Miller–Madow bias correction (Miller, 1955). Results are robust to using MI redundancy and Jeffreys-smoothed probabilities.

For our second analysis we perform causal mediation analysis (Imai et al., 2010) to link our interventions to performance, via multi-agent synergy as a mechanism. We estimate the model using the `{mediation}` package in **R** (Tingley et al., 2014). We find that the ToM intervention has a significant indirect effect on performance through its impact on synergy, beyond any direct treatment effects.

## A.12 ROBUSTNESS TESTS USING "EARLY SYNERGY"

Since our experiment is censored when groups correctly guess the target number, length of data in our analyses varies between groups. As robustness test, we also compute "early synergy" using only data up to round 15 (removing about 40% of the data). Using the practical criterion, we find about 5% of individual $p$-values are below 0.05. Fisher test is highly significant overall ($p = 2.7 \times 10^{-6}$), significant in the Plain (0.001) and ToM condition ($p = 0.0004$) and marginally significant in the Persona condition ($p = 0.059$). Using a more aggressive cutoff in round 10 (dropping only about 20% of trials but giving multi-agent systems less time to evolve synergy and having less data for estimation we find only significant in the ToM condition ($p = 0.028$).

## A.13 EXPERIMENTS WITH OTHER MODELS

To test generalizability of our findings, we conducted experiments using four other models:

1. LLAMA 3.1 8B,
2. LLAMA 3.1 70B INSTRUCT,
3. Google GEMINI 2.0 FLASH, and
4. QWEN3 235B A22B INSTRUCT 2507.

We conducted 100 replications of the group experiment with ten agents and temperature of 1.0 of each model and each treatment condition.

Before turning to comparative results, we first address a structural challenge that emerged with QWEN3—one that speaks to a broader issue with deploying reasoning models in multi-agent coordination settings.

---

[5]As measures for task difficulty we use (a) how far groups target number was from the mid point (smaller numbers requiring more steps of the binary search process); and (b) the modulo remainder of the target number and the group size (target numbers that are closer to divisibility by the group size are easier to guess by pure chance).

[6]While the raw outcome is binary and binomial regression would be appropriate, after the IPW weighting step, outcomes are no longer binary, hence quasibinomial model.

As a reasoning model, QWEN3 exhibited persistent looping behavior during its chain-of-thought process—a known failure mode of reasoning models (Pipis et al., 2025; Cemri et al., 2025; Shapira et al., 2026). These models are particularly susceptible to self-reflection and deductive reasoning traps (Baek & Tegmark, 2025) and circular reasoning (Duan et al., 2026). In our setting, we identify three specific patterns of oscillatory reasoning that arise reliably as soon as group feedback becomes inconsistent with the individual binary search strategy.

**Adjacent Integers.** When group feedback places an agent between two adjacent integers, the agent fails to recognize that the ambiguity of that feedback is driven by the unobservability of other agents' guesses. A typical reasoning pattern was: "8 was too low but 9 was too high, but guesses need to be integers, let me re-read the instructions ..."). The agent interprets the goal literally and realizes they are mutually incompatible, not realizing that changes in the guesses of other agents explain the seeming inconsistent feedback.

**Dual Strategies.** Agents fail to reconcile the strategy inconsistency of both reasoning about the task (applying binary search strategy) while also coordinating with other agents. A typical reasoning pattern was: "I should guess the midpoint 5 ... but I also need to adjust to others ... so maybe I should guess higher ... but that's not the midpoint ... let me re-read the instructions ...".

**Mutual Mental Modeling.** We observe four different categories of mutual mental modeling traps, each representing a distinct cognitive stance.

1. Uncertainty Acknowledgment: "Perhaps the other players guess is changing. But no information." Agents recognize that other agents may be changing their behavior and admit epistemic helplessness. This shows awareness of others, without actual modeling.
2. Causal Attribution Under Ambiguity: "Perhaps the mystery number is for the sum, and when you guess 10, the sum was too high, but maybe the other guesses changed." Here, the agent is using mental modeling to interpret the group feedback. This shows mutual modeling as a diagnostic tool to explain the inconsistency.
3. Strategic Simplification: "Given the instruction to use binary search, I think we must assume others are constant." This is a meta-level decision to collapse the modeling problem by deliberately treating other agents as static to make the task tractable. In some sense, this is the most sophisticated move as it recognizes that mutual modeling creates infinite regress and chooses a pragmatic simplification.
4. Concrete Joint-State Estimation: "I should guess 9, and others 8 — sum 17. So guess 9. Or if others are at 8, I should guess 9. But I don't know." This is an actual attempt to simulate the joint outcome by assigning specific values to other agents' guesses yet it dissolves into uncertainty.

These examples illustrate that the model is aware of the task ambiguity and often correctly recognizing the oscillating pattern (often, reasoning trace includes the word phase "[we are] oscillating"). That is, the failure is not due to reasoning error. Instead it is failure to terminate under irreducible epistemic uncertainty about others. We term this *paralysis under coordination ambiguity*. Notably, the paralysis persists even at a high temperature setting ($T = 1.0$), which should increase stochastic exploration and disrupt deterministic decoding loops, indicating that the instability arises from internal reasoning dynamics rather than sampling artifacts. As a practical remedy, we managed to improve robustness by adding one extra line to the prompt to to the QWEN experiments:

> If you find yourself re-evaluating your guess more than once, repeat your last guess.

Overall, we find evidence of multi-agent LLM system capacity for emergence in other models, with some variation across models with different capacity and treatment condition (Table A2). We find the task is quite hard for LLAMA 3.1 8B, and the majority of groups fail to solve it. In the vast majority of instances, groups get stuck in oscillating behavior of guessing too high (low) and then overcompensating by guessing too low (high), unable to break out of the cycle by establishing specialized differentiation across agents and pursuing complementary behavior. We find evidence of emergence using emergence capacity test (10.5% of individual groups show $p$-values are below 0.05; the joint Fisher test is highly significant). Robustness tests using bias corrected estimates on time-trend residualized data find significant signs of emergence in the **Persona** condition but marginal and

no evidence in the two other conditions. The test for emergence using the practical criterion shows no evidence for emergence (only 3.3% of individual groups show $p$-value below 0.05 and joint Fisher suggests no evidence with $p = 1$ but the robustness test using the bias-corrected estimates on time-trend demeaned data suggest emergence in the **Plain** condition ($p = 0.011$), no evidence in the **Persona** condition ($p = 0.232$), and marginal evidence in the **ToM** condition ($p = 0.068$).

Groups of LLAMA 3.1 70B agents perform substantially better. We find strong evidence of emergence using the emergence capacity criterion (both overall and in the persona condition). All tests with the practical criterion are significant. Groups of GEMINI 2.0 FLASH agents perform even higher. We find strong evidence of emergence using the emergence capacity criterion (overall Fisher test is significant, as well es significant robustness test in the **Plain** and **ToM** condition). Performance of groups of QWEN3 agents falls between LLAMA 70B and GEMINI 2.0 FLASH. We find strong evidence of emergence using both the emergence capacity criterion and the practical criterion with most tests showing highly significant $p$-values (only the Wilcoxon test in the **Plain** is marginal at 0.052). The insignificant $I_3$ in groups of QWEN agents could be a result of the failure modes outlined above: the reasoning process of alignment with other agents is never successful given the paralysis under coordination ambiguity. The insignificant $G_3$ in groups of LLAMA 3.1 8B agents could be a result of that model's generally much lower ToM ability.

| | Llama-3.1-8B | Llama-3.1-70B | Gemini 2.0 flash | Qwen3 |
|---|---|---|---|---|
| **Group Performance** | | | | |
| $\text{Success}_{Plain}$ | 11% | 53% | 60% | 51% |
| $\text{Success}_{Persona}$ | 14% | 59% | 75% | 71% |
| $\text{Success}_{ToM}$ | 5.5% | 61% | 71% | 58% |
| **Emergence Capacity** | | | | |
| Fisher | 0.000 | 0.000 | 0.000 | 0.000 |
| $\text{Wilcoxon}_{Plain}$ | 0.252 | 0.110 | 0.006 | $1.7 \times 10^{-10}$ |
| $\text{Wilcoxon}_{Persona}$ | 0.020 | 0.012 | 0.193 | $2.4 \times 10^{-11}$ |
| $\text{Wilcoxon}_{ToM}$ | 0.061 | 0.200 | 0.003 | $9.5 \times 10^{-11}$ |
| **Practical Criteria** | | | | |
| Fisher | 1.000 | 0.015 | 0.662 | $2.4 \times 10^{-4}$ |
| $\text{Wilcoxon}_{Plain}$ | 0.011 | 0.002 | 0.215 | 0.052 |
| $\text{Wilcoxon}_{Persona}$ | 0.233 | 0.001 | $3.2 \times 10^{-5}$ | 0.010 |
| $\text{Wilcoxon}_{ToM}$ | 0.068 | 0.006 | $4.9 \times 10^{-6}$ | $2.1 \times 10^{-4}$ |
| **Groups with Significant Differentiation** | | | | |
| Plain | 86% | 68% | 64% | 89% |
| Persona | 92% | 77% | 77% | 94% |
| ToM | 80% | 86% | 86% | 97% |
| **$I_3$** | | | | |
| Fisher | 0.000 | 0.000 | 0.000 | 1.000 |
| **$G_3$** | | | | |
| Fisher | 1.000 | 0.000 | 0.000 | 0.000 |

Table A2: Experiment results with other LLM models.

### A.14 SPECIFIC PERSONA EFFECTS

Both the **Persona** and **ToM** conditions rely on personas. Do the specific personas—or combinations of personas—present in a multi-agent system affect performance or the emergence of higher-order structures? We explore these questions with several test. Recall that our experiments are based on 20 different personas that are generated once globally, and then reused throughout the experiments by sampling 10 personas from the library (without replacement). First, we perform joint F-tests (based on linear regression) of the null hypothesis that all persona dummies have no effect on the outcome (i.e., no specific persona substantially affects group outcomes and all coefficients equal zero). Rejecting the null would indicate that at least one of the persona indicators is significantly

associated with the outcome variable. We find no evidence that the presence of specific personas effect performance ($F = 1.217$; $p = 0.239$), number of rounds to success ($F = 1.356$; $p = 0.144$), PID synergy ($F = 0.839$; $p = 0.66$), or PID redundancy ($F = 0.596$; $p = 0.910$). These results suggest that what matters is the presence of personas overall—markers that allow the LLM agents to center their differentiation on—rather than who those personas are specifically.

Next, we used an LMRA (Eloundou et al., 2024; Riedl & Weidmann, 2025) to judge the pairwise similarity of the personas in terms of their personality and background (we used `gpt-oss-120b` to assess similarity on a 0-10 scale). Even though judgments are subjective, the input they judge is standardized, controlled, information-rich, and deliberately designed to reveal deep-level characteristics in line with personas used in vignette studies (Aguinis & Bradley, 2014). This allows the LMRA to integrate many deep-level characteristics in a controlled and comparable format. From these, we compute group-level diversity as one minus the mean pairwise similarity rating, following the dispersion-based operationalization of separation diversity in Harrison & Klein (2007) (mean diversity: 5.457; SD = 0.385). In the last step, we explore whether group diversity is systematically related with key group-level outcomes using linear regression. We find no effect of diversity on performance ($\beta = 0.067$; $p = 0.403$), number of rounds to success ($\beta = -0.195$; $p = 0.877$), PID synergy ($\beta = -0.000$; $p = 0.856$), or PID redundancy ($\beta = 0.003$; $p = 0.113$). These results again suggest that there are no strong effects for specific combinations of personas.

