# OpenReview forum: "Emergent Coordination in Multi-Agent Language Models"
_ICLR.cc/2026/Conference — ICLR 2026 Poster_

### Official Review · Reviewer_st12 · 2025-10-29

**Soundness:** 4
**Presentation:** 3
**Contribution:** 3
**Rating:** 6
**Confidence:** 3

**Summary:**

The recent advancements in large language models have showcased the effectiveness of multi-agent systems. However, the authors argue that the performance gains in these multi-agent systems are not well understood, beyond the emergent behavior due to from interacting specialist agents. The core motivation is then to determine the threshold when a population of LLM agents transfers from a collection of agents into a productive collective. The authors investigate a group guessing game where agents must propose integers such that all integers sum a hidden unknown target, where the only feedback is binary high/low of the group sum. For this task to be successful, members of the group must fulfill complementary roles in an emergent way. The authors formulate three heuristics to measure the group synergy and emergence with the goal of quantifiability: practical criterion (prediction of delayed-release macro signal from instantaneous agent states), emergence capacity (median synergy estimate across all agent pairs), and coalition test (difference in predictive ability of a triplet compared to its most predictive pair). Each heuristic covers some needed insight that may not be available by the other heuristics.

The heuristics are used across two tests alongside null distribution: shuffling row wise to break agent identities, and shuffling column-wise to break time alignment. The authors propose a separate series of hierarchical models to differentiate agents across rounds. The main intervention in the paper is changing the text prompt input of each LLM agent, using either plain prompts (treating LLMs as human subjects), persona prompts (instructing each agent to respond conditional on anthropomorphic attributes), and theory of mind prompts (instructing agents to act based on what they perceive other agents to think). Empirical experiments using GPT4.1 and Llama-3.1-8b first showed that practical criterion for emergence is satisfied regardless of intervention. Next the hierarchical model showed that the most differentiation among agents occurs for Persona and ToM prompting. The coalition test is used to show that ToM agents tend towards higher task mutual information while coalition signal was lower, suggesting that among pairs of agents in the ToM set, there is a reinforcement of established roles with respect to the group goal. This is in contrast to the Persona set, which has a similar total mutual information to Plain (Figure 3b). A subset of experiments were repeated on Llama-3.1-8b but had mixed results due to the reduced generalization ability.

**Strengths:**

- The authors investigate the problem of emergent synergy in LLM-based agents, which is relevant to the NLP and multi-agent research communities, yet difficult to quantify. Tools from information theory and partial information decomposition are used to propose three heuristics.
- The heuristics provide measurement tools to study task alignment among triads and pairs of agents, alongside the grounding of agent micro-states in the delayed system signal (practical criterion). These heuristics could have significance for the broader research community in studying emergent synergy of multi-agent systems.
- The authors provide confidence values and error margin for each reported result. The plot rendering is generally high quality with only minor issues in some figures. The writing quality is high enough to convey the technical points.
- The empirical analysis reveals some interesting insights about tuning LLM prompts to use ideas from ToM with GPT 4.1. These ideas do not necessarily transfer over to smaller models such as LLama-8b. An interesting takeaway from the analyses was that ToM prompts may reinforce pairwise identity assignments given to agents. The pairwise reinforcement, qualitatively speaking, may be reinforced first by the original system prompt of an agent, and again by the other agents projecting the identity onto the same agent during ToM reasoning.
- For the group sum task, it is shown that triadic synergy may be spurious compared to strictly pair-wise synergy, in fact it may be detrimental as evidenced by Persona and ToM coalition measurements.

**Weaknesses:**

- The writing in some sections could be clarified to help motivate heuristics and models. For example "Test of Agent Differentiation" mixed model is not well motivated as a heuristic until reaching the results in Section 4.2, some sign posting would help clarify the utility.
- Some conceptual details are not clear or self contained, such as the author's use of PID (L157), it is not clear how unique information and redundant information are formulated for the group sum task.
- The notation needs some clarification to make the proposed formulation self-contained: the variables $t$, $l$, $k$ (L138-L144) should be introduced more naturally, and choice of $devs$ for target deviations may be changed to a single bold-face letter.
- Typos: L335-L344 Figure 3 is referenced as Figure 4, L480 emergency -> emergent
- Vertical figure subtitles are difficult to read and tend to clash with the figure identifier.

**Questions:**

- Considering the lower $G_3$ of ToM agents in Figure 3c, do the authors still see utility in the coalition test heuristic as a measure of the system's emergent synergy? It would seem that simply using the pairwise mutual information says more about the system capability than a triad approach in the group sum setting. In other words, for the specific task of group sum, triadic mutual information does not appear necessary for task success since Persona and Plain score higher, so it may be misleading to label triadic synergy as "dynamical emergent synergy" when pairwise synergy suffices.

---

> ### Author Response · Authors · 2025-11-21
> **Re: Utility of G3 as measure of the system's emergent synergy => localize synergy, rather than treat it as sign of system capacity overall**
>
> Thank you for the thoughtful comment. We agree it’s important to disentangle what each statistic measures and what it contributes. You are right, the ToM condition shows lower G3 (coalition gain) but higher I3 (total triadic TDMI). This is exactly the redundancy-synergy trade‑off we emphasize: the ToM condition produce stronger alignment on the macro signal (high I3), such that any pair already carries most of the macro‑relevant information (hence lower G3). By contrast, higher G3 in Plain/Persona often reflects that no pair suffices because the macro signal is fragmented, not that the system is more capable. We have made this distinction more explicitly throughout, and especially in the introduction (e.g., by adding a new panel to Figure 1a). Furthermore, we have added new analyses that tie synergy and redundancy directly to performance (Section Results -> Increased Performance). In particular, we find that individually synergy and redundancy have negative effects on performance, but when both are present the effect on performance is positive. We see this as exactly the strength of the paper: not only do we explore emergence capacity in multi-agent systems, but localize it, and tie it to performance. In that way, lower G3 in the ToM condition is not a sign of “lower performance” but rather that synergy needs to be goal directed and aligned to be effective.
>
> High pairwise MI can be driven by redundancy or by temporal autocorrelation; it does not tell us whether complementary interactions are present or whether beyond‑pair structure is needed. The coalition test answers a different question: is there macro‑relevant information that only emerges when we consider three agents together? Low G3 under ToM is consistent with better alignment (any pair nearly suffices), not with a lack of capability.
>
> The coalition test remains useful precisely because it localizes where macro predictability depends on beyond‑pair structure. It complements the other two tests
> pairwise PID synergy to a joint future establishes existence of order‑2 synergy; and
> the macro criterion, which screens for beyond‑univariate predictive structure.
> Taken together, the three tests shed light on the coordination regime: ToM shifts groups toward high macro alignment with limited need for beyond‑pair coalitions; Plain/Persona rely more on idiosyncratic triadic configurations.
>
> In summary, we see the coalition test as a diagnostic for whether the macro signal requires beyond‑pair structure, not as a necessary condition for success. The ToM condition’s pattern (higher I3 and lower G3) indicates better goal alignment with less reliance on fragile triadic configurations, which aligns with its superior performance. This adds additional explanatory insight that sharpens the interpretation of the synergy-redundancy tradeoff of the PID decomposition tests. Thank you for these suggestions. We have made several edits throughout the paper to clarify why and how the four different tests complement each other.

---

> ### Author Response · Authors · 2025-11-21
> **Re: writing and notation improvements**
>
> Thank you so much for reading our paper so carefully and offering such detailed and constructive suggestions for improvement. We have fixed the typos, figure reference, and revised the writing to address the points you raise as well as several others to improve readability (such as adding numbers to our equations for easier reference).

---

> ### Author Response · Authors · 2025-11-27
>
> We would like to thank the reviewer again for their feedback. We wanted to check whether our response addresses your concerns. We're happy to provide further clarification if needed.

---

### Official Review · Reviewer_EcL2 · 2025-11-01

**Soundness:** 3
**Presentation:** 3
**Contribution:** 2
**Rating:** 4
**Confidence:** 3

**Summary:**

This paper investigates whether multi-agent LLM systems exhibit emergent collective properties beyond individual agents. The authors develop an information-theoretic framework using partial information decomposition and time-delayed mutual information to quantify synergy in multi-agent systems. They test this framework on a number guessing game under three conditions: plain instructions, assigned personas, and theory-of-mind prompting. Results show all conditions exhibit statistical synergy, with different coordination patterns across interventions. The ToM condition displays higher mutual information and lower emergent synergy compared to other conditions.

**Strengths:**

S1. The paper addresses the important and novel question of when multi-agent systems exhibit collective intelligence that transcends the sum of individual agents, which is crucial for exploring MAS capabilities and LLM potential.

S2. The authors adapted an appropriate experimental paradigm from cognitive science work (the group guessing game), which inherently requires collective group interaction.

S3. ToM prompting induces identity-locked differentiation and goal-directed complementarity, which to some extent distinguishes whether synergy is genuinely functional.

S4. The experimental design incorporates robust validity checks and null hypotheses, such as employing multiple entropy estimators and redundancy measures in partial information decomposition (PID). Additionally, the sample size is sufficient to ensure statistical significance.

**Weaknesses:**

W1. The experiments may have limitations in causal inference. For instance, the Persona and ToM interventions simultaneously alter multiple factors. Designing separate control groups to isolate these two factors could better explain the causal relationships in the experiments.

W2. Figure 2(a) shows no statistically significant differences in success rates across different interventions, suggesting that ToM may not necessarily be optimal, and that the effectiveness of synergy requires stronger evidence. Under ToM intervention, I3 is higher than G3. High I3 indicates that agent behaviors contain substantial information about group goals, but it remains unclear whether this information is redundant or complementary. G3 indicates whether observations of agents can be decomposed into pairwise relationships. The paper explains these results as distinguishing effective from spurious synergy, with low G3 being a sign of alignment. However, this does not rule out the phenomenon of apparent alignment due to excessive agent homogenization rather than better coordination. Furthermore, high I3 does not lead to better task performance. I3 may only measure information quantity rather than quality, or the task itself may not require complex synergy to solve. Therefore, I believe ToM may indeed change the interaction patterns between agents, but further validation is needed regarding I3's predictive power and whether ToM can be claimed as a superior mode.

W3. The term "emergence" may be overused in this paper. Synergy does not necessarily represent the emergence of collective intelligence; it merely indicates that the joint predictive ability of agents exceeds the sum of their individual predictive abilities, which does not imply any cooperation or specialized roles. The paper is potentially imprecise in claiming specialized roles, using hierarchical models showing significant agent-level random intercepts and G3>0 as evidence. However, this cannot necessarily be attributed to roles but may stem from temperature randomness or prompt differences in the persona condition. It is difficult to determine whether this represents emergent behavior or merely the result of prompt engineering.

**Questions:**

C1. Quantile binning with K=2 may lose critical dynamic information by binarizing continuous guess values. Moreover, under K=3 bins, the ToM condition shows p=0.991 for emergence capacity, and the paper does not explain this problematic value.

---

> ### Author Response · Authors · 2025-11-18
> **re: is ToM intervention necessarily helping performance or not? => yes, it does! new analyses added**
>
> This is such a great point! We agree with you completely that linking emergent synergy to performance is a crucial step that was missing from our analysis. The pattern of results around I3 and G3 pointed in that direction but wasn’t fully able to establish what is going on and whether ToM is ultimately helpful or not. We have now added detailed analyses linking emergent synergy with performance (Section What does Emergence Allow? Paragraph “Increased Performance”). Here we use regression analysis and causal mediation to show that emergent behavior does indeed lead to higher performance and we have some evidence that the ToM intervention is a driver of it.
>
> Specifically, we find that on their own, higher levels of either synergy or redundancy do not predict success. However, when both are present, performance improves significantly (significant interaction with $\beta = 0.24$; $p = 0.014$). In marginal-effect terms, redundancy amplifies the benefit of synergy on the log-odds scale by 27\%; and vice versa synergy amplifies benefits of redundancy by 27\%. This pattern implies that systems benefit when redundant pathways create goal alignment, while synergistic interactions extract novel, non-overlapping information---together enabling higher overall performance.
>
> We complement the regression analysis with causal mediation analysis (Imai et al, 2010). While reaching only marginal significant levels the effect is consistent: the ToM treatment causally increases performance indirectly by increasing synergy (ACME $= 0.034$ [95\%CI: $-0.000 - 0.07$], $p = 0.053$). This aligns with the interpretation that performance benefits emerge when systems achieve both redundancy (aligned toward a common goal) and synergistic integration (differentiated, complementary roles)---a form of functional and collective complexity (Varley et al., 2023; Luppi et al., 2024; Sterelny et al. 2007).
>
> Together, these new results (combined with the other evidence in the paper and summarized well by your comment) provide much stronger evidence that the ToM intervention itself is responsible for performance differences.

---

> ### Author Response · Authors · 2025-11-18
> **Re: Synergy does not necessarily represent the emergence of collective intelligence => more clarity and new analyses provide stronger evidence**
>
> Thank you for this critical dissection of the pattern of results. We have added two sets of analyses to make this clearer and alleviate the concerns you mention. First, we have added additional analyses linking our TDMI-PID quantities (synergy and redundancy) to performance. This strengthens our claim that synergy is indeed performance relevant and not just an artifact. Second, we have some evidence that the ToM condition indirectly affects performance by increasing synergy via causal mediation analysis (ACME $= 0.034$ [95\%CI: $-0.000 - 0.07$], $p = 0.053$; see previous response for more details. Third, we have added new analyses to explore whether persona content and diversity affect observed group-level outcomes (Appendix “Specific Persona Effects”). We find no effect of specific personas or group diversity on (a) performance, (b) number of rounds, or (c) emergent higher-order structures (PID redundancy and synergy; see new Appendix Section Specific Persona Effects). Finally, you are right that temperature induced randomness can equally “seed” identities which can then serve as the basis for stable between agent differentiation. This is exactly why we observe emergent synergy even in the Plain condition. However, random temperature based variation provides only a weak signal.

---

> ### Author Response · Authors · 2025-11-21
> **re: C1. Quantile binning with K=2 vs K=3 => new analyses and explanation added**
>
> Thank you for this great comment. We agree that coarse binning can discard signals which is exactly why we added the K=3 binning robustness test. The downside of this is that with just around ~10 observations, K > 2 rapidly induces sparse contingency tables and high-variance estimates. You are right to point to the high p-value for the ToM model in the K = 3 case which we did not discuss. Here is what we think is going on with some additional results that support this interpretation.
>
> Agents in the ToM condition behave in a more structured way (see evidence on shared goal alignment with the much higher mutual information I3). That is, the data is most “polarized”, splitting the observations more extremely across bins. As a result, many joint cells become rare or empty, inflating estimator variance and making permutation tests conservative (biasing against finding synergy if it is present). The effect (more empty/sparse cells) is stronger in the ToM condition (which has more structure, which concentrates probability mass on fewer outcomes, leading to more polarized bin counts) compared to the Plain and Persona condition (which has less structured interactions, more diffuse bin counts), producing conservative p-values in our permutation test. The high p-value reported in the K = 3 with standard time-trend demeaning should be taken to reflect estimator variance and a wide distribution of null values, not evidence of absence of emergence.
>
> To corroborate this we look at the same analysis but use a different demeaning strategy. Namely, we take the functional null model residuals (functional baseline without between-agent synergy). I.e., we take the same approach as shown in Column 3 of Table A1, but compute them with K = 3 instead of K = 2.
>
> Plain: mean -0.001; Wilcoxon p-value = 0.999
> Persona: mean = 0.0003; Wilcoxon p-value = 0.896
> ToM: mean = 0.0014; Wilcoxon p-value = 0.025*
>
> Here, the pattern reverses and the ToM condition shows significant evidence of emergence. The functional null residual is subtracting out the “easy” structure (shared trends, individual autocorrelation, and correlated responses to the group-level feedback) that masquerades as synergy in the Plain and Persona conditions. However, the null can’t subtract the complementary, cross-agent structure that is actually present in ToM (because the functional baseline is without between-agent synergy). Using this form of residualizing, the ToM still has systematic joint predictability beyond the naïve baseline and thus exceeds the permutation null. In the ToM condition, role-like complementarity is not captured by the naïve model and remains, even in the K = 3. Here, residualization recenters/compresses distributions, improving bin occupancy,  and stabilizes the discrete TDMI calculation and the sparsity penalty from the simpler time-trend demeaning weakens specifically for ToM. This alternative approach to residualizing sharpens the null and rebalances occupancy, so only genuinely complementary, cross-agent structure remains.
>
> In summary, a stricter test isolates exactly the construct of interest—synergy beyond coordination-free dynamics—and ToM is the only condition that passes it. The K=3 failure for the simpler time-trend demeaning is a methodological sensitivity (sparsity and a fat null), whereas the residualized success is substantive (effect persists after removing easy structure). Together they show the ToM effect is robust to deconfounding but fragile under known low-power discretization—precisely why we pre-specify K=2 for discrete PID and use residualized/continuous checks as corroboration.
>
> We will add these new results and a summary of this discussion to the K = 3-bins robustness section in the appendix. Thank you for suggesting it.

---

> > ### Comment · Reviewer_EcL2 · 2025-11-26
> > **Official Comment by Reviewer EcL2**
> >
> > The responses addressed most of my questions. I would like to raise my score to 6.

---

### Official Review · Reviewer_Hc1h · 2025-11-03

**Soundness:** 3
**Presentation:** 3
**Contribution:** 3
**Rating:** 4
**Confidence:** 4

**Summary:**

This paper proposes a framework to test whether multi-agent LLM systems exhibit emergent coordination and higher-order structure (above and beyond mere collections of individuals). The authors apply partial information decomposition (PID) of time-delayed mutual information (TDMI) to measure dynamical emergence in multi-agent systems solving a group guessing game. Through experiments with GPT-4.1 and Llama-3.1-8B agents under three prompt conditions (control, persona assignment, and personas with theory of mind instructions), they demonstrate that multi-agent LLM systems can be steered from simple aggregates to integrated collectives exhibiting goal-directed synergy.

**Strengths:**

To my knowlege, this work is the first systematic application of information theoretical (PID/TDMI) framework to LLM multi-agent systems. I really like prompt manipulations to demonstrate that coordination patterns can be influenced by prompt engineering. Multiple permutation tests and bias correction methods have been leveraged.

**Weaknesses:**

1. While the emergent properties of LLM collective groups are well-characterized through TDMI, it is less clear "why" they show these patterns of emergent coordination. Is it because they actually lead to performance gains? (i.e., they coordinate because that's actually helpful to find the solution) Or do they simply emerge from mimicking human behavioral patterns in their training data, regardless of performance benefits? Figure 2 shows no clear performance differences between Plain, Persona, and Persona + ToM conditions, and the paper does not directly show LLM performance with vs. without emergent coordination, so it's hard to assess whether their emergent coordination really drives the performance.

2. The abstract claims LLM coordination patterns "mirror well-established principles of collective intelligence in human groups," so I hoped to see more comparisons between human vs. LLMs. Without human baseline data (or cited literature) using the same TDMI measures, it was hard to assess whether LLM collectives are more, less, or equally emergent compared to humans.

3. Single task, dramatic model-dependency (GPT-4.1 vs Llama-3.1 show opposite patterns)

**Questions:**

1. I hope to see several methodological details:
- Were the same 20 personas reused across 200 experiments or regenerated by GPT?
- How were N=10 personas selected for agents in each run? (randomly selected? fixed? or based on certain criteria?)
- Could you provide full examples of actual personas provided? (full 20 persona)
- These ambiguities affect interpretation: are effects (i.e., "persona" effects) due to having "distinct identities" or "specific persona combinations"? (i.e., is the effect driven by having "any distinct persona" or having "certain set of persona"?)

2. The role of "ToM prompt" is a bit unclear to me. The task inherently requires considering others' contributions (summing to target). What does the ToM prompt add beyond this implicit requirement?

---

> ### Author Response · Authors · 2025-11-18
> **Re: Additional method details re personas => robustness tests added**
>
> Were the same 20 personas reused across 200 experiments or regenerated by GPT?
> => yes, we generated 20 personas once and reused them across all experiments.
>
> How were N=10 personas selected for agents in each run? (randomly selected? fixed? or based on certain criteria?)
> => for each group-experiment we randomly sampled 10 personas from the fixed set (without replacement).
>
> Could you provide full examples of actual personas provided? (full 20 persona)
> => yes, absolutely. You can find them here (blind link on OSF): https://osf.io/8z7b5/overview?view_only=ac79ff0d072e4240ac019a3273d9426a
> We will add the personas to the GitRepo of the final paper as well.
>
> These ambiguities affect interpretation: are effects (i.e., "persona" effects) due to having "distinct identities" or "specific persona combinations"? (i.e., is the effect driven by having "any distinct persona" or having "certain set of persona"?)
>
> This was such a great suggestion! We performed several sensitivity tests to explore whether any of the effects are driven by the presence of a specific persona, or the combination (diversity) of personas. TL;DR we find no effect of specific personas or group diversity on (a) performance, (b) number of rounds, or (c) emergent higher-order structures (PID redundancy and synergy). These new results are now part of the Appendix Section Specific Persona Effects.
>
> **== new analyses added ==**
>
> First, we perform joint F-tests (based on linear regression) of the null hypothesis that all persona dummies have no effect on the outcome (i.e., no specific persona substantially affects group outcomes and all coefficients equal zero). Rejecting the null would indicate that at least one of the persona indicators is significantly associated with the outcome variable. We find no evidence that the presence of specific personas effect performance ($F = 1.217$; $p = 0.239$), number of rounds to success ($F = 1.356$; $p = 0.144$), PID synergy ($F = 0.839$; $p = 0.66$), or PID redundancy ($F = 0.596$; $p = 0.910$). These results suggest that what matters is the presence of personas overall---markers that allow the LLM agents to center their differentiation on---rather than who those personas are specifically.
>
> Next, we used an LMRA (Eloundou et al. 2024) to judge the pairwise similarity of the personas in terms of their personality and background (we used gpt-oss-120b to assess similarity on a 0-10 scale). Even though judgments are subjective, the input they judge is standardized, controlled, information-rich, and deliberately designed to reveal deep-level characteristics in line with personas used in vignette studies (Aguinis et al. 2014). This allows the LMRA to integrate many deep-level characteristics in a controlled and comparable format. From these, we compute group-level diversity as one minus the mean pairwise similarity rating, following the dispersion-based operationalization of separation diversity in (Harrison & Klein 2007; mean diversity: 5.457; SD = 0.385). In the last step, we explore whether group diversity is systematically related with key group-level outcomes using linear regression. We find no effect of diversity on performance ($\beta = 0.067$; $p = 0.403$), number of rounds to success ($\beta = -0.195$; $p = 0.877$), PID synergy ($\beta = -0.000$; $p = 0.856$), or PID redundancy ($\beta = 0.003$; $p = 0.113$). These results again suggest that there are no strong effects for specific combinations of personas.

---

> ### Author Response · Authors · 2025-11-18
> **re: role of "ToM prompt" is a bit unclear to me => it helps agents recognize oscillation**
>
> You are right: it seems a bit counter intuitive that the ToM prompt would have any meaningful effect since it is already implicitly required to solve the task. The contribution stems from the fact that agents need to trade off two different strategies. On the one hand, a standard group binary search (GBS) process (splitting the difference between the two known bounds to hone in on the target value). On the other hand, they need to consider other agents when the GBS process alone is not sufficient and the group oscillates. Consider the case where a group of 10 agents needs to guess the mystery number 55. If every agent guesses 5 the sum is 50, which is too low. If every agent consequently increases their individual guesses by the minimum amount to 6, the sum is 60, which is too high. To break out of this oscillation, agents need to realize that the target number may be between the two previous guesses; they need to explicitly consider that “if we’re all switching from 5 to 6 we might over-shoot”. Without the ToM prompt, agents fail to do this reliably and oscillate more. Hence, it’s not so much that ToM isn’t already implied by the task itself, but LLM agents fail to consider it reliably if it is absent from the prompt instructions.

---

> ### Author Response · Authors · 2025-11-18
> **Re: does emergent coordination drive performance? => yes it does! new analyses added**
>
> This is such a great point! We agree with you completely that linking emergent synergy to performance is a crucial step that was missing from our analysis. We have now added detailed analyses linking emergent synergy with performance (Section What does Emergence Allow? Paragraph “Increased Performance”). Here we explore regression analysis and causal mediation to that emergent behavior does indeed lead to higher performance.
>
> Specifically, we find that on their own, higher levels of either synergy or redundancy do not predict success. However, when both are present, performance improves significantly (significant interaction with $\beta = 0.24$; $p = 0.014$). In marginal-effect terms, redundancy amplifies the benefit of synergy on the log-odds scale by 27\%; and vice versa synergy amplifies benefits of redundancy by 27\%.
>
> This pattern implies that systems benefit when redundant pathways create goal alignment, while synergistic interactions extract novel, non-overlapping information---together enabling higher overall performance. We complement the regression analysis with causal mediation analysis (Imai et al, 2010). While reaching only marginal significant levels the effect is consistent: the ToM treatment causally increases performance indirectly by increasing synergy (ACME $= 0.034$ [95\%CI: $-0.000 - 0.07$], $p = 0.053$). This aligns with the interpretation that performance benefits emerge when systems achieve both redundancy (aligned toward a common goal) and synergistic integration (differentiated, complementary roles)---a form of functional and collective complexity (Varley et al., 2023; Luppi et al., 2024; Sterelny et al. 2007).

---

> ### Author Response · Authors · 2025-11-18
> **Re: Abstract claims of specialization—coordination tradeoff => more clarification added**
>
> Thank you for the suggestion. The statement was meant to draw a parallel on the construct / theory level, not at the metric or magnitude level (whether multi-agent LLMs are more, less, or equally emergent compared to humans). We claim that LLM collectives exhibit patterns consistent with foundational team-science principles—namely, alignment on shared objectives and complementary, differentiated contributions—not that they match humans on any particular scale. This tradeoff is a well-established regularity across the team literature, although no one has used that same TDMI measure to allow direct comparison. These principles are among the most replicated findings in team research across decades, tasks, and measures (e.g., shared goals/mental models; transactive memory/role specialization; tradeoff between positive effects of diversity on group memory and negative effects on team conflict). In fact, a popular definition of “teams” even refers to individuals with “complementary skills who are committed to a common purpose” (p.8, Katzenbach & Smith, 1993). Given this breadth, construct-level correspondence does not require reuse of a specific instrument to be justified.
>
> The new analyses we added regarding the joint effect of redundancy & synergy (see above) now clarify this connection more explicitly. Individually, synergy & redundancy have negative effects on performance, but when both are present, the effect is positive. This is exactly the pattern we draw a parallel to when making this connection. We have added additional clarifications (e.g., Figure 1a) and citations to the discussion.
>
> Interestingly, this same tradeoff also seems to be central for Mixture of Experts (MoE) models— a case where multi-agent systems exist within a single model. Here, achieving high specialization requires carefully balancing expert utilization via routing capacity constraints. More recent work (e.g., Expert-Choice Routing by Zhou et al. 2022) explicitly frames the problem as a trade-off: if specialization is too weak, experts collapse (=> there is no synergy); but if routing tries to enforce perfect balance, capacity constraints and communication overhead degrade model performance (=> alignment is hard to achieve). Thus, MoEs must negotiate a fundamental specialization–coordination tradeoff. Our work provides a theoretical and methodological foundation for how to quantify and assess this tradeoff.

---

> ### Author Response · Authors · 2025-11-27
>
> We would like to thank the reviewer again for their feedback. We wanted to check whether our response addresses your concerns. We're happy to provide further clarification if needed.

---

### Official Review · Reviewer_A7pR · 2025-11-05

**Soundness:** 2
**Presentation:** 2
**Contribution:** 2
**Rating:** 6
**Confidence:** 4

**Summary:**

This paper explores when multi-agent LLM systems exhibit genuine higher-order structure—that is, coordination dynamics that go beyond mere aggregation of individual behaviors. The authors introduce an information-theoretic framework to test for dynamical emergence and cross-agent synergy using partial information decomposition (PID) and time-delayed mutual information (TDMI). They formalize three complementary tests—practical, emergence capacity, and coalition tests—to detect and localize synergy among agents in data-driven experiments. The empirical analysis uses GPT-4.1 and LLaMA-3 agents playing a “group guessing” task without explicit communication, under various prompting conditions (Control, Persona, ToM). The results show that only the Theory-of-Mind (ToM) condition leads to consistent “goal-directed” synergy and identity-linked differentiation across agents. The findings connect principles of collective intelligence from human groups to emergent behavior in LLM collectives.

**Strengths:**

The paper tackles a timely and underexplored question—whether multi-agent LLM collectives can exhibit emergent higher-order coordination—through a rigorous, information-theoretic lens.
The proposed framework is conceptually elegant and connects human-group cognition with quantitative, falsifiable tests for emergence. The use of partial information decomposition and time-delayed mutual information is particularly novel in the context of LLM collectives, moving beyond superficial behavioral correlations. The three-layered test design (practical, emergence capacity, coalition) provides complementary diagnostics and robustness checks. The authors are careful to isolate emergent coordination from spurious synchronization or heterogeneity-induced artifacts via well-chosen surrogate null models.

The experimental setup is simple but powerful: a communication-free group guessing game allows the authors to observe spontaneous coordination. The finding that prompting agents to adopt Theory-of-Mind reasoning produces structured, goal-aligned differentiation is compelling and resonates with the literature on human collective cognition. Overall, the work offers a rigorous empirical entry point to the study of collective behavior in AI systems, a topic of growing significance for multi-agent AI safety and orchestration.

**Weaknesses:**

While the empirical results are intriguing, the theoretical underpinnings of the framework could be deepened. The notion of “emergence” here is operationalized through information decomposition, but the connection between these information-theoretic quantities and game-theoretic or dynamical systems perspectives remains underdeveloped. For instance, one might ask:
	•	Can the framework be linked to known fixed-point or equilibrium concepts (e.g., correlated equilibria, variational stability) to characterize steady-state synergy?
	•	Is the “macro signal” in their tests analogous to a coarse-grained order parameter in dynamical systems theory, and could Lyapunov-like quantities be defined to formalize stability of emergent coordination?
	•	How does the introduced TDMI-based synergy relate to mutual predictability under stochastic gradient dynamics or belief-updating processes?
Developing a mathematical bridge to these established theories could significantly increase the conceptual depth of the paper.

In addition, the paper’s experimental validation—while extensive—focuses on a single synthetic task. It would be valuable to test whether the same synergy measures predict performance or alignment quality in richer multi-agent environments (e.g., collaborative reasoning or negotiation tasks). The robustness of the results to changes in LLM temperature, prompt diversity, and task complexity could also be more systematically analyzed.

Finally, while the authors emphasize “no communication,” there remains an implicit shared context via prompts and the task description, which could bias coordination outcomes. A discussion on how to distinguish true emergent coordination from shared prompt priors would improve the interpretability of results.

**Questions:**

ΝΑ

---

> ### Author Response · Authors · 2025-11-21
> **re: Extensive validation on a single task => new analyses offer more details linking synergy to performance**
>
> Thank you---we agree that evaluating only one task is a limitation. We chose the group-sum setting deliberately because it is the minimal environment that instantiates the redundancy-synergy tension very cleanly (alignment vs complementarity), admits clear null models (row vs column surrogates), and avoids confounds from tool use or rich communication. This lets us validate the methodology itself (PID on TDMI, localization via surrogates and coalition test), setting the stage for future work to move to more complex domains. Within this controlled setting, we already stress-tested across two model families (GPT‑4.1, Llama‑3.1‑8B), prompt regimes (Plain/Persona/ToM), a grid over group size and temperature, two entropy corrections, and multiple falsification baselines---so our emphasis here is depth on a canonical task rather than breadth over tasks.
>
> That said, we agree with you that establishing a more concrete link between emergent synergy and performance is a crucial step. To that end, we have now added detailed analyses linking emergent synergy with performance (Section What does Emergence Allow? Paragraph “Increased Performance”). Here, we use regression analysis and causal mediation to explore how emergent behavior does indeed lead to higher performance.
>
> Specifically, we find that on their own, higher levels of either synergy or redundancy do not predict success. However, when both are present, performance improves significantly (significant interaction with $\beta = 0.24$; $p = 0.014$). In marginal-effect terms, redundancy amplifies the benefit of synergy on the log-odds scale by 27\%; and vice versa synergy amplifies benefits of redundancy by 27\%.
>
> This pattern implies that systems benefit when redundant pathways create goal alignment, while synergistic interactions extract novel, non-overlapping information---together enabling higher overall performance. We complement the regression analysis with causal mediation analysis (Imai et al, 2010). While reaching only marginal significant levels the effect is consistent: the ToM treatment causally increases performance indirectly by increasing synergy (ACME $= 0.034$ [95\%CI: $-0.000 - 0.07$], $p = 0.053$). This aligns with the interpretation that performance benefits emerge when systems achieve both redundancy (aligned toward a common goal) and synergistic integration (differentiated, complementary roles)---a form of functional and collective complexity (Varley et al., 2023; Luppi et al., 2024; Sterelny et al. 2007).

---

> ### Author Response · Authors · 2025-11-21
> **Re: W3 - could implicit shared context via prompts and the task description be a confound?**
>
> Thank you for raising this---distinguishing true “emergence” from effects induced by shared instructions or global feedback is exactly what our nulls are designed to test. We construct surrogates and a functional null that preserve the shared context while stripping cross-agent dependence (the row-wise shuffles, the column-wise shuffles, and the functional null. Residualizing the multi-agent data with this functional null baseline explicitly “partials out” the common prompt/task description and the global response to feedback, leaving no cross-agent channel. Any synergy that remains, is the “true emergent coordination” you refer to. Empirically, our results then show that shared prompt priors and group feedback are not sufficient to explain the observed structure. In short, we fully agree that shared prompts and global feedback create the potential for correlated behavior as a baseline. Our surrogates, bias correction, and functional nulls are designed to tease out synergy beyond that baseline. The residual, significantly positive, bias-corrected synergy and alignment metrics are precisely the “true emergent coordination” you ask us to distinguish. Furthermore, this is also precisely why we focus on a single, canonical task, where we can develop such a comprehensive set of surrogates and functional null models: disentangling true emergent coordination from shared prompt priors requires task‑tailored functional nulls and surrogate controls. This would be difficult to design and validate across heterogeneous tasks within one paper. Depth on a single controlled setting lets us build and validate these diagnostics rigorously, so they can later be transferred to richer multi‑agent environments.

---

> ### Author Response · Authors · 2025-11-23
> **re: Game theory and complex systems ties => new analyses added and paper restructured**
>
> We thank the reviewer for this insightful comment. It prompted us to significantly deepen our theoretical analysis by explicitly linking our information-theoretic framework to dynamical systems stability and game-theoretic equilibria. To address this, we introduced new diagnostic measures derived from our TDMI framework: Total Stability (I3 normalized by the entropy of the macro-signal). This provides us an "Order Parameter" of the predictability of the macro-scale outcome given the micro-scale agent states, serving as a proxy for the depth of the system's basin of attraction (as with our other analyses, use rigorous modeling with time-trend demanding and null-model bias correction). This now yields a nuanced picture of multi-agent coordination in form of a transition to stability. As you hypothesized, we observed a clear dynamical shift. Control groups exhibited chaotic dynamics (Stability around 0), while the ToM intervention acted as a control parameter, driving the system into a stable steady-state (Total Stability significantly > 0). Building further on complex systems theory, we can now better explain the nature of the equilibrium. While the system is "emergent" (the group possesses information the individuals do not), decomposing this into information gain at the triplet level (over information provided by pairs alone) we find low G3. This suggests that alignment happens on the dyad level, rather than on the triplet level. This result maps directly to the specific constraints of our task. Because agents receive only global feedback (and cannot observe other agents' individual actions), they cannot align with specific local partners but must couple with the "Mean Field" of the group. Each agent treats the “rest of the group” as a single partner and cannot form ties with specific individuals. This new metric allows us to answer your question regarding the nature of the equilibrium. The system does not settle into a fragile, high-complexity state (which would be indicated by high G3). Instead, the ToM prompt steers the system into a robust, redundant correlated equilibrium where stability is maintained by dense pairwise alignment with the global feedback signal. This mirrors "mean field" dynamics in statistical physics, where global coupling creates ordered phases without requiring complex local rules.
>
> We have restructured the paper along three pillars:
> 1. Existence of Synergy (analysis of synergy capacity and practical emergence criterion).
> 2. Dynamical Mechanism
>     - ToM acts as control parameter moving system from chaos to stability
>     - Using coalition decomposition (G3), we show this stability is driven by dense pairwise alignment (via Mean Field) rather than complex triplet locks.
>     - Identity Anchoring: This alignment is not transient noise; Personas provide the stable fixed points (random intercepts) that allow this pairwise structure to lock in.
> 3. Functional consequences (also new): emergence is not just statistical curiosity but synergy & redundancy are drivers of group performance.

---

> ### Author Response · Authors · 2025-11-27
>
> We would like to thank the reviewer again for their feedback. We wanted to check whether our response addresses your concerns. We're happy to provide further clarification if needed.

---

### Author Response · Authors · 2025-11-23

We thank all reviewers for their thoughtful and constructive feedback. We have incorporated your suggestions into the revised manuscript and uploaded an updated version. Your recommendations have significantly strengthened the work. Most notably, new results (described below) show that the **emergent synergy is tied to higher performance**.

We summarize our clarifications and updates below:

 - **New analyses demonstrate functional usefulness of synergy for the task:**
Several of you suggested that a more concrete link between emergent synergy and performance would help strengthen our results. To that end, we have now added detailed analyses linking emergent synergy with performance (Section What does Emergence Allow? Paragraph “Increased Performance”). Here, we use regression and causal mediation analysis to explore how emergent behavior does indeed lead to higher performance. We sharpened the conceptual framework to explicitly include functional usefulness as a new pillar. We have re-arranged some sections to highlight that no single metric spans all goals; instead: the strength arises from the combination of all our analysis.
    - Pillar 1: Demonstrate emergence capacity
    - Pillar 2: Dynamical mechanism (basin of attraction, fixed points, equilibrium behavior, identity-linked differentiation)
    - Pillar 3: Functional consequences (higher performance)

- **Increased conceptual depth of the paper by developing connections with game theory and complex systems:**
We significantly deepen our theoretical analysis by explicitly linking our information-theoretic framework to dynamical systems stability and game-theoretic equilibria (new Lyapunov-like quantity as proxy for the depth of the system's basin of attraction and mean field like coupling creates ordered phases). This offers an expanded interpretation of our theory of mind (ToM) findings: the ToM intervention acts as control parameter that shifts multi-agent LLM interaction dynamics from chaotic to stable.

 - **Expanded robustness tests with k=3 binning, B=2000 random shuffled baselines:**
These added robustness tests allow us to better distinguish true “emergence” from what would be expected by shared context or global feedback alone. Overall, they strengthen the conclusion of true “emergence”.

 - **Expect more rigorous and transparent statistical validation and presentation:**
We expanded and enhanced the presentation of our various tests (explicit condition-wise p-values, Wilcoxon tests, conservative demeaning, enumerated robustness checks) and clearer descriptions of data, nulls, and formalisms. We also further developed our conceptual framework regarding the specialization vs. integration tradeoff.

 - **New analyses demonstrate that personas matter overall, not the presence of any specific persona or diversity of personas in the multi-agent system:**
New results show that no specific persona matters for performance (or emergence); neither does the diversity of personas. Personas function more as a “coordination device” rather than anchors of specific behavior.

We really want to acknowledge and show our deep appreciation of the reviewers’ insights and believe that these revisions meaningfully strengthen the clarity, transparency, and soundness of the work.

---

### Meta-Review · Area_Chair_JeR9 · 2026-01-13

**Summary:**

The paper introduces an information-theoretic, data-driven framework to assess when multi-agent LLM systems transfer from a collection of agents into a integrated collective with higher-order structure. The authors implement both a practical criterion and an emergence capacity criterion operationalized as partial information decomposition of time-delayed mutual information (TDMI). The authors test a group guessing game where agents propose integers such that all integers sum a hidden unknown target and the feedback is group-level  high/low of the sum. This task tests whether the agents can form complementary roles in an emergent way. The authors formulate three heuristics to measure the group synergy and emergence including practical criterion, emergence capacity, and coalition test. Experiments with GPT-4.1 and Llama-3.1-8B agents under three prompt conditions (control, persona assignment, and personas with theory of mind/TOM instructions) show that multi-agent LLM systems can be steered from simple aggregates to integrated collectives exhibiting goal-directed synergy.

The reviewers all appreciate the problem being timely and underexplored. The proposed heuristics provide measurement tools to study task alignment and would be useful for the broader research community in studying emergent synergy of multi-agent LLMs. The reviewers also found the findings from experiments insightful. The main concerns including the emergent synergy not clearly tied to performance (Reviewer Hc1h, A7pR, EcL2), limiting to a single synthetic task (Reviewer A7pR, Hc1h), connection to known fixed-point or equilibrium concepts (Reviewer A7pR), strong model dependency where GPT-4.1 and Llama-3.1-8B show opposite behavior (Reviewer Hc1h, st12). The authors clarified the questions and provided new experiments result to address the concerns, which made the claims solid. There are some remaining concerns such as limitation in single task and model dependency. Overall this is a borderline paper. Considering the importance of the problems and potential impact of proposed measuring heuristics, the recommendation is accept. The authors should revise the paper substantially in next version to incorporate the response and consider addressing the outstanding concerns.

**Reviewer Concerns:**

Addressed concerns:

Emergent synergy not clearly tied to performance (Reviewer Hc1h, A7pR, EcL2): This is a shared concern. The authors provided new experiments and analysis, and argued ToM improves performance indirectly through synergy.

Connection to known fixed-point or equilibrium concepts (Reviewer A7pR): the authors added new analysis that explicitly linking information-theoretic framework to dynamical systems stability and game-theoretic equilibria, which made the claims stronger.

Outstanding:

Limiting to a single synthetic task (Reviewer A7pR, Hc1h): Generalization to more multi-agent settings remains open.

Model dependency (Reviewer Hc1h, st12): strong model dependency where GPT-4.1 and Llama-3.1-8B show opposite behavior. The authors discussed the differences, yet broader cross-model evidence would clarify the question.

**Reviewer Scores:**

Reviewer EcL2 (4->6): Explicitly stated to raise score to 6.
Reviewer A7pR (6), Reviewer st12 (6): The reviewers are both positive about the paper and likely remain positive.
Reviewer Hc1h (4): The reviewer  raised issues on performance linkage, human comparison, and model dependence; performance linkage and persona-method details were substantially addressed, so a modest upward shift is plausible.

---

### Decision · Program_Chairs · 2026-01-26

Accept (Poster)